# How to Fine-Tune a Reasoning Model? A Teacher–Student Cooperation Framework to Synthesize Student-Consistent SFT Data

**Zixian Huang** [1]   **Kaichen Yang** [1 2]   **Xu Huang** [3]   **Feiyang Hao** [1]   **Qiming Ge** [1]   **Bowen Li** [1]   **He Du** [1]   **Kai Chen** [1]   **Qipeng Guo** [1]

## Abstract

A widely adopted strategy for model enhancement is to use synthetic data generated by a stronger model for supervised fine-tuning (SFT). However, for emerging reasoning models like Qwen3-8B, this approach often fails to improve reasoning capabilities and can even lead to a substantial drop in performance. In this work, we identify substantial stylistic divergence between teacher generated data and the distribution of student as a major factor impacting SFT. To bridge this gap, we propose a Teacher–Student Cooperation Data Synthesis framework (TESSY), which interleaves teacher and student models to alternately generate style and non-style tokens. Consequently, TESSY produces synthetic sequences that inherit the advanced reasoning capabilities of the teacher while maintaining stylistic consistency with the distribution of the student. In experiments on code generation using GPT-OSS-120B as the teacher, fine-tuning Qwen3-8B on teacher-generated data leads to performance drops of 3.25% on LiveCodeBench-Pro and 10.02% on OJBench, whereas TESSY achieves improvements of 11.25% and 6.68%. Our code is available at https://github.com/CoopReason/TESSY.

## 1. Introduction

Recently, major AI companies have increasingly introduced reasoning models as flagship large language models (LLMs) (Jaech et al., 2024; Guo et al., 2025a; Yang

[1]Shanghai AI Laboratory [2]Dalian University of Technology [3]Nanjing University. Correspondence to: Zixian Huang <huangzixian@pjlab.org.cn>, Qipeng Guo <guoqipeng@pjlab.org.cn>.

*Proceedings of the 43$^{rd}$ International Conference on Machine Learning*, Seoul, South Korea. PMLR 306, 2026. Copyright 2026 by the author(s).

et al., 2025). These models generate responses that bifurcate into a thinking content for fine-grained reasoning and a final answer content that delivers the solution. By introducing explicit thinking processes, LLMs have achieved new breakthroughs in reasoning-intensive tasks such as code generation (Sun et al., 2024; Zheng et al., 2025; Wang et al., 2025).

Given the varying capabilities and inference costs of different-sized models, a common approach involves large models generating SFT training data for smaller models, allowing them to achieve similar task performance (Ahmad et al., 2025; Guha et al., 2025; Liu et al., 2025). However, although reasoning models often have higher baseline performance, such work is still primarily carried out on Base or Instruct models, with few attempts on reasoning models (Yang et al., 2024a; Hui et al., 2024).

A key factor limiting further SFT of reasoning models is that higher-quality but distributionally different data can easily cause catastrophic forgetting (Luo et al., 2025a; Chen et al., 2025a). However, open-source reasoning models are typically fine-tuned in-house on large, proprietary datasets, the specific composition and distribution of which are not publicly disclosed. As a result, using new synthetic data without sufficient knowledge of the original data distribution risks introducing subtle but harmful distributional conflicts, which can in turn degrade overall model performance.

The problem is further exacerbated by the stylistic differences among reasoning models. As illustrated in Figure 1, although the code-proficient GPT-OSS-120B (Agarwal et al., 2025) produces the correct solution strategies, the style of its connecting text differs noticeably from that of Qwen3-8B (Yang et al., 2025). Directly using data generated by GPT-OSS-120B to train Qwen3-8B would force the latter to make unnecessary adaptations, which is undesirable and may even be detrimental. Moreover, previous studies indicate that even models of the same family on different scales can exhibit stylistic differences (Chen et al., 2025b).

To address this challenge, in this paper, we aim to inves-

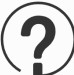

Given a string s, you can delete any palindromic substring in one operation. Find the minimum number of operations to remove the whole string.

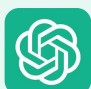

**Teacher: GPT-OSS-120B**

**We need to solve a problem ...**
**We can consider removing s[i] alone: dp[i][j]=1+dp[i+1][j] ...**
**We can try to pair s[i] with later s[k]: dp[i][j] = min(dp[i][j], dp[i+1][k−1] + dp[k+1][j]) ...**

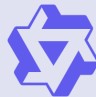

**Student: Qwen3-8B**

**Okay, let's see. The problem is to ...**
**For example, if s[i] == s[j], then perhaps dp[i][j] = min(dp[i][j−1], dp[i+1][j], dp[i+1][j−1] + 0) ...**
**Alternatively, here's an idea: dp[i][j] could be min(dp[i][j−1], dp[i+1][j], dp[i+1][j−1]) ...**

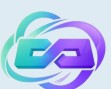

**Teacher-Student Cooperation Data Synthesis**
GPT-OSS-120B + Qwen3-8B

**Okay, let's see. The problem is to ...**
**For example, removing s[i] alone: dp[i][j]=1+dp[i+1][j] ...**
**Alternatively, here's an idea: pair s[i] with later s[k]: dp[i][j] = min(dp[i][j], dp[i+1][k−1] + dp[k+1][j]) ...**

*Figure 1.* Illustration of the Teacher–Student Cooperation Data Synthesis Framework. **Blue** and **Pink** denote the capability-related text spans generated in the responses of GPT-OSS-120B and Qwen3-8B, respectively. **Green** and **Purple** indicate the stylistic text segments produced by GPT-OSS-120B and Qwen3-8B. The objective of our synthesis framework is to delegate the generation of capability-related text to GPT-OSS-120B, while assigning the generation of stylistic text to the student model Qwen3-8B.

tigate *how to synthesize data that preserve the reasoning capabilities of the teacher model while maintaining consistency with the distribution of the student model.* As illustrated in the bottom of Figure 1, our goal is to generate responses in which the solution-relevant text is produced by the teacher model, while the remaining stylistic text is generated by the student model. To achieve this, we propose a **Te**acher–**S**tudent Cooperation Data **Sy**nthesis framework (**TESSY**), which alternately uses student and teacher models to generate stylistic content and reasoning content, respectively. This division of labor preserves the accuracy of the reasoning and alleviates SFT interference arising from differences in text style. To implement this division precisely, TESSY uses a generate-then-rollback strategy, allowing fine-grained control over the responsibilities of the teacher and student models.

Experiments on code generation task shows that directly performing SFT with teacher-only data, using GPT-OSS-120B as the teacher to train the stylistically distinct student Qwen3-8B, results in performance drops of up to 3.25% and 10.02% in LiveCodeBench-Pro (Zheng et al., 2025) and OJBench (Wang et al., 2025), respectively. In contrast, TESSY increases the performance of Qwen3-8B by 11.25% and 6.68% on the same benchmarks. Further experiments show that TESSY delivers consistent gains when paired with different teacher models, including DeepSeek-R1 and the student-related Qwen3-235B-A22B-Thinking.

## 2. Related Work

**Reasoning Model.** The LLMs paradigm has shifted to models with deep reasoning capabilities, using extended reasoning traces to decompose complex tasks (Jaech et al.,

2024). Although SFT on synthetic data is a proven strategy for imparting capabilities to base or instruct models (Guo et al., 2025a; Yang et al., 2025), it remains underexplored how to further refine a reasoning model, especially since the original training recipes are largely undisclosed. In addition to this, recent studies indicate that reasoning models exhibit distinct styles in different organizations and scales (Guo et al., 2025b; Chen et al., 2025b). Moreover, while tokens like "Wait" and "Hmm" are conventionally regarded as mere stylistic markers, emerging evidence suggests that they play a functional role in facilitating complex reasoning (Goyal et al., 2024; Qian et al., 2025). These insights suggest that the SFT data should be carefully designed to preserve the intrinsic reasoning style of the model (Li et al., 2025b).

**Catastrophic Forgetting.** Catastrophic forgetting has long been a fundamental challenge (Kirkpatrick et al., 2017; Li & Hoiem, 2017), and currently preserving the knowledge acquired during the earlier training stages has become a central concern (Luo et al., 2025b). One line of work mitigates it by constraining parameter updates. Parameter-efficient fine-tuning, which modifies only a small subset of parameters, has become a practical choice for large models (Hu et al.; Jiang et al., 2024). However, freezing most parameters also limits the model's capacity to acquire new information (Biderman et al.). Another line of work emphasizes the distribution of training data. Previous studies suggest that learning from the on-policy data sampled from the student model can reduce forgetting by maintaining distributional consistency (Shenfeld et al., 2025; Chen et al., 2025a). However, using only data generated by student can underutilize teacher knowledge. To balance knowledge retention and transfer, we propose a teacher–student

framework that generates student-aligned data enriched with teacher knowledge.

**Teacher-Student Collaboration.** Training student models using data synthesized by a teacher is a widely adopted approach (Kim & Rush, 2016; Lin et al., 2020). However, due to differences in capacity and style, a stronger teacher does not necessarily produce a stronger student (Li et al., 2025a). Some methods adapt the teacher to a student-friendly distribution, but retraining teachers with hundreds of billions of parameters is often prohibitively expensive (Park et al., 2021; Sengupta et al., 2024). Self-distillation mitigates the distribution mismatch by having the student rewrite teacher-generated references (Yang et al., 2024b; Gupta et al., 2025), but this process can introduce shortcut behaviors in reasoning models. The closest works are on-policy distillation (Agarwal et al.), where the teacher supervises student-generated data, and methods Speculative Distillation and AdaSwitch further improve this via alternating generation (Xu et al., 2025; Peng et al., 2025). However, training models with different vocabularies remains a challenge. Moreover, unlike these online approaches, our work focuses on synthesizing offline SFT data that efficiently transfers knowledge across models and tasks, enabling practical and open use.

## 3. Approach

### 3.1. Research Objective

The goal of SFT is to align the model's output distribution with the data distribution. Let $P_D(y_i \mid x, y_{<i})$ denote the data distribution of the $i$-th token given the input $x$ and the previous tokens $y_{<i}$, and let $P_{\mathcal{M}_S}(y_i \mid x, y_{<i})$ denote the predictive distribution induced by the student model $\mathcal{M}_S$. For notational simplicity, we omit the explicit conditioning on $x$ and $y_{<i}$ throughout the paper, and denote the corresponding conditional distributions by $P_D(y_i)$ and $P_{\mathcal{M}_S}(y_i)$. Under this notation, the training objective is defined as

$$\mathcal{L}(\mathcal{M}_S) = \sum_{i=1}^{n} \mathbb{E}\Big[\mathrm{KL}\big(P_D(y_i) \parallel P_{\mathcal{M}_S}(y_i)\big)\Big], \quad (1)$$

where $n$ denotes the total number of tokens. For $P_D(y_i)$, an efficient approach is to synthesize it based on a more powerful teacher model $\mathcal{M}_T$.

We further consider the output sequence as consisting of two types of tokens. Let $\mathcal{T}_{\mathrm{Cap}}$ denote the index set of capability tokens that are directly related to task solving, such as code or numerical tokens, and $\mathcal{T}_{\mathrm{Sty}}$ denote the index set of style tokens that are task-irrelevant, such as tokens expressing tone or discourse (e.g., "wait", "but"). Consequently, the

training objective can be decomposed as

$$\mathcal{L}(\mathcal{M}_S) = \mathcal{L}_{\mathrm{Cap}}(\mathcal{M}_S) + \mathcal{L}_{\mathrm{Sty}}(\mathcal{M}_S),$$

$$\text{where} \quad \mathcal{L}_{\mathrm{Cap}}(\mathcal{M}_S) = \sum_{i \in \mathcal{T}_{\mathrm{Cap}}} \mathbb{E}\Big[\mathrm{KL}\big(P_D(y_i) \parallel P_{\mathcal{M}_S}(y_i)\big)\Big],$$

$$\mathcal{L}_{\mathrm{Sty}}(\mathcal{M}_S) = \sum_{i \in \mathcal{T}_{\mathrm{Sty}}} \mathbb{E}\Big[\mathrm{KL}\big(P_D(y_i) \parallel P_{\mathcal{M}_S}(y_i)\big)\Big].$$

$$(2)$$

To improve performance on the target task, our primary objective is to optimize the loss related to capability $\mathcal{L}_{\mathrm{Cap}}(\mathcal{M}_S)$. As for the style-related loss $\mathcal{L}_{\mathrm{Sty}}(\mathcal{M}_S)$, although it does not directly contribute to the enhancement of task performance, learning it is generally harmless in conventional settings where a base version model is fine-tuned, and is often relatively easy to learn. Consequently, previous works have not explicitly distinguished the style loss.

However, for reasoning models, both the teacher and student models have already undergone extensive pretraining and thus exhibit distinctive stylistic patterns. The training setups of each model are different, leading to substantial stylistic discrepancies between the teacher and the student. This issue is even more pronounced when the teacher and the student come from different sources. As a result, directly optimizing $\mathcal{L}_{\mathrm{Sty}}(\mathcal{M}_S)$ becomes challenging and can even adversely affect the learning of $\mathcal{L}_{\mathrm{Cap}}(\mathcal{M}_S)$.

To mitigate the influence of style tokens $\mathcal{T}_{\mathrm{Sty}}$ and ensure sufficient learning of capability tokens, our objective is to construct synthetic data such that the distribution of capability tokens is sampled from the teacher model, while style tokens are encouraged to follow the distribution of the student model. Formally, we construct the synthetic data by generating each token $y_i$ as follows:

$$y_i \sim \begin{cases} P_{\mathcal{M}_T}(\cdot), & \text{if } i \in \mathcal{T}_{\mathrm{Cap}}, \\ P_{\mathcal{M}_S}(\cdot), & \text{if } i \in \mathcal{T}_{\mathrm{Sty}}. \end{cases} \quad (3)$$

To achieve the objective in Equation (3), we introduce the teacher–student cooperation data synthesis framework (TESSY) in the following.

### 3.2. Teacher–Student Cooperation

As illustrated in Figure 2, TESSY generates a response in an alternating and iterative manner between the teacher and the student models. In the following, we present the key design components of TESSY.

**Alternating Generation** TESSY aims to generate capability tokens using the teacher model to ensure correctness, while using the student model to produce style tokens that align with the student's data distribution. During the human

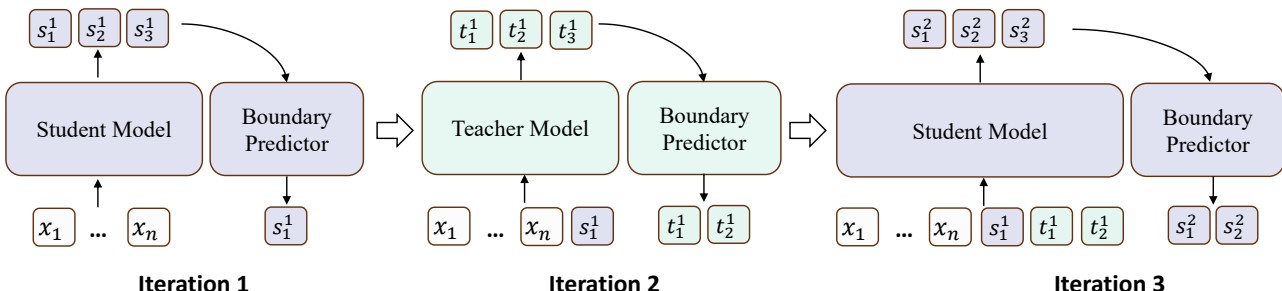

*Figure 2.* Overview of the teacher–student cooperation data synthesis framework. The teacher and student models alternately generate capability and style tokens, while the boundary predictors enforce truncation to the exact capability or style spans.

reasoning process, reasoning steps are often connected by transitional or connective statements. Accordingly, in the reasoning trajectory generated by the model, capability and style tokens appear alternately. Therefore, for a synthetic sample $y$, TESSY generates capability and style spans by alternating between teacher and student models:

$$y = [s^1, t^1, s^2, t^2, \dots]$$

where $s^i$ and $t^i$ denote the spans of multiple tokens generated by the student and the teacher, respectively. In practice, because model output often begins with a stylistic phrase such as "Okay, let's see," TESSY starts the generation process with the student model producing the first span $s^1$.

For each span $s^i$ and $t^i$, generation depends on all previously generated spans and is defined as

$$s^i = \mathcal{M}_S\Big(x, [s^1, t^1, \dots, s^{i-1}, t^{i-1}]\Big),$$
$$t^i = \mathcal{M}_T\Big(x, [s^1, t^1, \dots, s^i]\Big). \tag{4}$$

However, a key challenge for TESSY is the generation boundary problem, which involves deciding the appropriate length of each span so that student-generated spans include only style tokens and teacher-generated spans include only capability tokens.

**Generation Rollback**  To address the generation boundary problem, we adopt a generate-then-rollback strategy. Specifically, at each step, the model first generates a fixed number of $k$ tokens, after which a boundary predictor is applied to determine the appropriate boundary and discard tokens beyond it.

Concretely, for the teacher model, we train a capability token boundary predictor to identify the position of the last capability token in the generated sequence and retain this token along with all preceding tokens to ensure that the resulting span contains only capability tokens. Similarly, for the student model, we train a style token boundary predictor to locate the last style token in the generated output and

retain this token along with all preceding tokens so that the resulting span consists purely of style tokens.

Formally, let $\tilde{s}^i$ and $\tilde{t}^i$ denote the raw spans generated by the student and teacher models, each consisting of a fixed number of $k$ tokens as described in Equation (4). We then apply the corresponding boundary predictors to truncate each span in a single step:

$$t^i = \tilde{t}^i_{<\mathcal{B}_{\mathrm{T}}(\tilde{t}^i)}, \quad s^i = \tilde{s}^i_{<\mathcal{B}_{\mathrm{S}}(\tilde{s}^i)} \tag{5}$$

where $\mathcal{B}_{\mathrm{T}}$ and $\mathcal{B}_{\mathrm{S}}$ denote the capability and style token boundary predictors for the teacher and student models, respectively.

Boundary predictors are implemented as token-level sequence labeling models similar to those used in traditional multi-span extraction tasks (Huang et al., 2023; 2024a), where a binary classification head is applied to each token to predict whether it belongs to a style or capability token. For the capability token boundary predictor, the boundary is defined as the first position predicted as a style token. For the style token boundary predictor, the boundary is defined as the first position predicted as a capability token.

To train $\mathcal{B}_{\mathrm{T}}$ and $\mathcal{B}_{\mathrm{S}}$, we randomly sample 100K segments of thinking content generated by the teacher and student models, respectively, and then prompt the teacher model to annotate all style spans in each segment. The details of the prompt can be found in Table 4. The boundary predictors are trained based on Qwen3-0.6B-Base, a sufficiently small model with a short input length, ensuring that their use does not compromise the efficiency of TESSY.

**Algorithm Details**  Based on the design principles outlined above, Algorithm 1 presents the complete data synthesis process of TESSY. TESSY begins by initializing generation with the student model in lines 5–6. In lines 7–20, the algorithm alternates between the teacher and student models in an iterative process.

During each iteration, TESSY follows a generate-then-rollback strategy (lines 8–11): the current generation model

**Algorithm 1** Teacher-Student Cooperation Data Synthesis

---

1: **Input:** prompt $x$, student model $\mathcal{M}_S$, teacher model $\mathcal{M}_T$, capability boundary predictor $\mathcal{B}_T$, style boundary predictor $\mathcal{B}_S$, maximum token number $k$
2: **Output:** Synthetic sequence $y$
3:
4: Initialize $y = [\,]$
5: Initialize generation model: $\mathcal{M} = \mathcal{M}_S$
6: Initialize boundary predictor: $\mathcal{B} = \mathcal{B}_S$
7: **while** not reaching final answer **do**
8:     Generate span: $\hat{z} = \mathcal{M}(x, y, k)$
9:     Predict boundary: $b = \mathcal{B}(\hat{z})$
10:     Truncate span: $z = \hat{z}_{<b}$
11:     Append $z$ to $y$
12:     **if** $\hat{z} \neq z$ **then**
13:         // Switch generation roles
14:         **if** $\mathcal{M}$ is $\mathcal{M}_S$ **then**
15:             $\mathcal{M} = \mathcal{M}_T, \mathcal{B} = \mathcal{B}_T$
16:         **else**
17:             $\mathcal{M} = \mathcal{M}_S, \mathcal{B} = \mathcal{B}_S$
18:         **end if**
19:     **end if**
20: **end while**
21: Generate final answer: $a = \mathcal{M}_S(x, y)$
22: Append $a$ to $y$
23: **Return** $y$

---

produces a span with a maximum token number $k$, which is then truncated by the corresponding boundary predictor to obtain a subsequence that aligns with the intended token type. This truncated span is appended to the synthetic sequence. Next, Line 12 checks whether a truncation has occurred. If truncation occurs, it indicates a change in the token type to be generated, necessitating a switch in the roles of the generation models. Lines 14–17 then switch to the generation model and its corresponding boundary predictor.

Finally, considering that the output of reasoning models is typically divided into a thinking part, which involves extensive intermediate reasoning, and a final answer, which is generally less complex and more stylistically distinct, TESSY delegates the generation of the final answer entirely to the student model. Therefore, once the thinking process is completed, TESSY exits the alternating generation loop and delegates the final answer generation to the student model (lines 21–22).

## 4. Experiments

### 4.1. Implementation Detail

For the implementation of TESSY, we empirically set the single-iteration maximum token number $k$ to 20. In cases where the student and teacher vocabularies differ, TESSY discards the last word to prevent semantic inconsistencies caused by subword mismatches. We implemented our framework based on vLLM (Kwon et al., 2023) and enabled prefix caching to support efficient model switching.

For the training setup, to avoid underestimating the baselines due to insufficient training, we trained all models for up to 9 epochs. All experiments were conducted using XTuner (Contributors, 2023b), with a batch size of 128 and a learning rate of 5e-5 on 32 H200 GPUs. During inference, we used the default maximum reasoning length of 40K for Qwen3-8B and set the temperature to 0.6.

### 4.2. Datasets

**Training Datasets** We collected open-source datasets released by OpenThoughts (Guha et al., 2025) and NVIDIA Nemotron (Liu et al., 2025), and used carefully designed prompts to guide GPT-OSS-120B (Agarwal et al., 2025) in selecting samples related to programming contest tasks. For each selected instance, we discarded the original response and retained only the corresponding question. From the remaining corpus, we randomly sampled 80K questions, comprising 37K unique questions, which were used to generate responses and to train models across all experiments.

**Evaluation Datasets** Our primary experiments focused on the code generation task, with mathematics and science question answering tasks used as auxiliary tests for out-of-domain evaluation. For code generation, we evaluated **LiveCodeBench-V5** (2024/08/01–2025/02/01) and **LiveCodeBench-V6** (2025/02/01–2025/05/01) (Jain et al., 2025), as well as **LiveCodeBench-Pro** (Zheng et al., 2025) and **OJBench** (Wang et al., 2025). For the auxiliary tasks, we evaluated **AIME-2024**, **AIME-2025**, and **Olympiad-Bench** (He et al., 2024) for mathematics, along with **GPQA-Diamond** (Rein et al., 2023) for science. For datasets containing multimodal data, only text-only parts were used.

All evaluations were conducted on the OpenCompass (Contributors, 2023a) platform, where pass@1 is computed as the average of results from multiple independent runs.

### 4.3. Models

We used GPT-OSS-120B (hereafter **GPT-OSS**) as the teacher model and **Qwen3-8B** as the student model, unless otherwise specified. In Section 5, we further evaluated the generalizability of our method by comparing alternative teacher models, including DeepSeek-R1-0528 (**DS-R1**) and Qwen3-235B-A22B-Thinking-2507 (**Qwen3-235B**), and an additional student model, Qwen3-30B-A3B.

*Table 1.* Performance comparison between TESSY and SFT baselines on in-domain code generation and out-of-domain test sets, where LiveCodeBench is abbreviated as LCB for brevity.

| | In-domain Test Sets | | | | Out-of-domain Test Sets | | | |
|---|---|---|---|---|---|---|---|---|
| | LCB-V5 | LCB-V6 | LCB-Pro | OJBench | GPQA-Diamond | AIME2024 | AIME2025 | OlympiadBench |
| Qwen3-8B | 55.09 | 49.58 | 25.35 | 18.75 | 60.16 | 76.67 | 69.17 | **69.81** |
| + Reject-Sampling | 54.55 | 48.00 | 26.06 | 18.32 | **60.61** | 75.42 | 66.77 | 69.53 |
| + Self-Distillation | 51.50 | 43.43 | 19.26 | 14.22 | **60.61** | 71.98 | 62.29 | 67.80 |
| + Teacher-Answer | 35.33 | 33.14 | 21.39 | 14.01 | 60.42 | 74.79 | 64.48 | 69.40 |
| + Teacher-Think | 47.90 | 43.43 | 24.36 | 12.50 | 54.29 | 77.81 | 69.06 | 69.59 |
| + Teacher-Mix | 45.51 | 40.57 | 21.95 | 11.85 | 57.39 | 75.21 | 61.56 | 68.60 |
| + Teacher-Only | 41.32 | 36.57 | 22.10 | 8.73 | 55.81 | 76.88 | 66.46 | 67.76 |
| + TESSY | **62.87** | **55.43** | **36.69** | **25.43** | 59.34 | **80.42** | **70.10** | 69.80 |

### 4.4. Compared Methods

We compared three categories of SFT data construction settings: student-driven, teacher–student collaborative, and teacher-driven data synthesis. The student-driven setting included **Self-Distillation**, where the teacher first generated a reference answer and the student generated the thinking and final answer based on the reference answer (Yang et al., 2024b), and **Reject-Sampling**, where the student generated multiple candidates and the teacher scored and selected the best one. In our experiments, five candidates were generated for selection. The teacher–student collaborative setting included **Teacher-Answer**, in which the student generated thinking content followed by the teacher producing the final answer, and **Teacher-Think**, which assigned thinking generation to the teacher and answer generation to the student. The teacher-driven setting included **Teacher-Only** (Kim & Rush, 2016), where all training samples were fully generated by the teacher, as well as **Teacher-Mix**, which mixed samples generated by the teacher and the student in a 1:1 ratio (Lin et al., 2020).

### 4.5. Experimental Results

Table 1 presents a comparison between TESSY and other SFT data synthesis approaches. Although training in synthetic data constructed by all other methods led to varying degrees of degradation in code generation performance, TESSY consistently improved Qwen-8B in all four datasets. Specifically, TESSY increased the performance of Qwen-8B by 7.78%, 5.85%, 11.34%, and 6.68% on the three Live-CodeBench datasets and OJBench, respectively. In contrast, only the Reject-Sampling method provided a marginal improvement of 0.71% on LiveCodeBench-Pro. Meanwhile, the widely used Teacher-Only approach, even after fully training up to 80K samples for 9 epochs, still suffered a performance drop of up to 10.02% in OJBench.

Observations on baselines indicate that, in general, the greater the teacher's participation in data synthesis, the greater the drop in student performance. For example, on OJBench, the fully teacher-generated method Teacher-Only caused a decrease of 10.02%. However, although reducing teacher involvement in data generation helps maintain model stability, it does not necessarily lead to performance gains, as exemplified by Reject-Sampling. Notably, although Self-Distillation is widely regarded as a promising approach to mitigate distribution mismatch in training data, our experiments reveal that it still incurs substantial performance degradation, consistent with the findings reported in (Kim et al., 2026). In contrast, TESSY, despite using up to 77.65% teacher-generated tokens as shown in Figure 20, did not cause any performance degradation and instead consistently improved the student model on in-domain datasets.

Furthermore, TESSY demonstrated potential for generalization to out-of-domain datasets. On AIME2024 and AIME2025, training with the data synthesized by TESSY improved Qwen3-8B by 3.75% and 0.93%, respectively, while preserving comparable performance on GPQA-Diamond and OlympiadBench. In contrast, other methods relying on teacher-generated data led to varying degrees of performance degradation. For example, Teacher-Only caused performance drops of 4.35%, 2.71%, and 2.05% on GPQA-Diamond, AIME2025, and OlympiadBench, respectively.

We have published the training data synthesized by TESSY [1].

## 5. Analysis

### 5.1. TESSY with Different Student Models

To evaluate the generality of TESSY, we further tested its effectiveness using the more capable MoE model Qwen3-30B-A3B as the student model. As shown in Figure 3, training Qwen3-30B-A3B with data constructed by TESSY led to performance improvements of 6.37% and 10.13% on LiveCodeBench-Pro and OJBench, respectively. While Qwen3-30B-A3B, likely due to its larger model capacity

---

[1] https://huggingface.co/datasets/CoopReason/TESSY-Code-80K

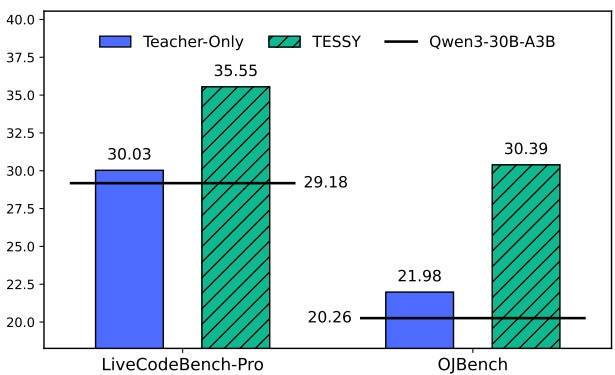

*Figure 3.* Experimental comparison using Qwen3-30B-A3B as the student model between Teacher-Only and TESSY.

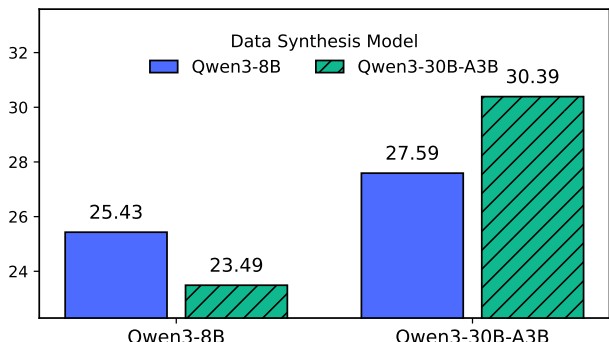

*Figure 4.* Performance of TESSY under different student models for data synthesis and model training on OJBench.

compared to Qwen3-8B, also benefited from training on Teacher-Only data, the gains were relatively modest. In contrast, TESSY provided additional improvements of 5.52% and 8.41% on the two datasets, demonstrating the ability of our method to generalize across different student models.

### 5.2. Importance of Consistent Student Models in Data Synthesis and Training

To further highlight the importance of aligning the data distribution with the target model during SFT, we conducted cross-training experiments in which data synthesized by Qwen3-8B was used to train Qwen3-30B-A3B, and vice versa. As shown in Figure 4, although Qwen3-30B-A3B exhibits stronger code generation capabilities overall, training Qwen3-8B on data generated by Qwen3-30B-A3B led to a 1.94% performance drop on OJBench compared to using data generated by Qwen3-8B. The effect was even more pronounced in the reverse setting: training Qwen3-30B-A3B on data generated by Qwen3-8B resulted in a 2.8% decrease. These results highlight that even within the same model family, mismatches between the training data distribution and the target model can significantly degrade performance during the SFT stage.

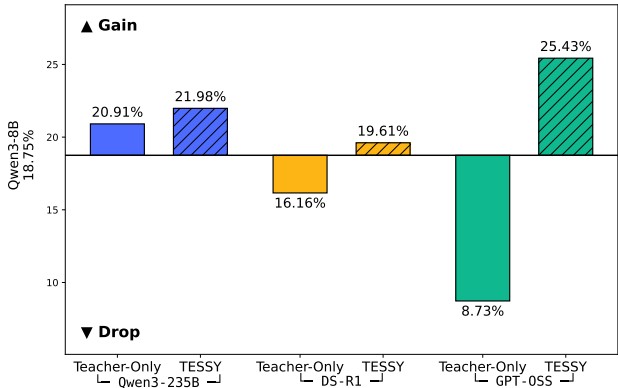

*Figure 5.* Comparison between Teacher-Only and TESSY with Qwen3-235B, DS-R1, and GPT-OSS-120B as teacher models.

### 5.3. TESSY with Different Teacher Models

Next, we equipped TESSY with different teacher models to further evaluate its generalization. As shown in Figure 5, even when using Qwen3-235B, a model of the same family as Qwen3-8B, TESSY outperformed Teacher-Only by 1.07%, demonstrating that distribution differences can have a significant impact on SFT performance. The effect of distribution mismatch was even more pronounced when using heterogeneous teacher models. In experiments with DS-R1 and GPT-OSS as teachers, Teacher-Only led to performance drops, while TESSY improved performance over Teacher-Only by 3.45% and 16.79%, respectively.

### 5.4. Quality of Synthesized Data

Unlike evaluating models trained with data synthesized by TESSY, we directly compared the responses generated by TESSY and Teacher-Only on OJBench in Figure 7 to assess the quality of the synthesized data. Although 22.4% of the tokens in TESSY were generated by the weaker Qwen3-8B model (as shown in Figure 20), TESSY still outperformed its teacher model, GPT-OSS-120B, by 10.99% when both were constrained to a maximum generation length of 40K tokens. Even when the maximum generation length of the teacher was increased to 48K, it remained 6.04% behind TESSY, and only slightly exceeded TESSY when extended further to 64K tokens.

These results suggest two key observations. First, incorporating the token generated by the student in TESSY does not significantly degrade generation quality, allowing effective use of the abilities of the teacher model. Second, the teacher's generation behavior in TESSY may be guided by the student, resulting in earlier termination of reasoning; in particular, the student's tendency to produce shorter thought traces may have guided GPT-OSS-120B towards earlier completion.

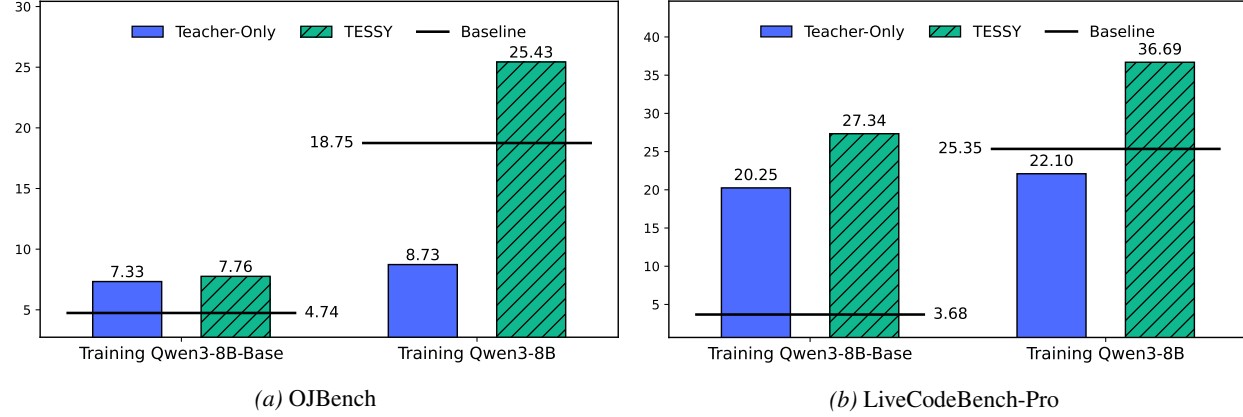

*(a)* OJBench

*(b)* LiveCodeBench-Pro

*Figure 6.* Comparison of TESSY and Teacher-Only across Base and Thinking versions of models.

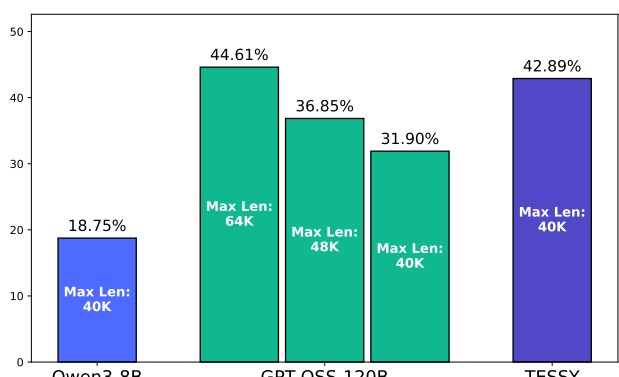

*Figure 7.* Data quality comparison: TESSY compared with direct data generation (Qwen3-8B, GPT-OSS) on OJBench.

### 5.5. Advantages of Training on Reasoning Model over Base Model

Given that SFT on reasoning models is inherently challenging, a natural question is *why not perform SFT starting from a more general base model instead?* To address this question, we compared the SFT performance of Qwen3-8B-Base and Qwen3-8B using the same training data synthesized by Teacher-Only and TESSY (with Qwen3-8B as the student model). As shown in Figure 6, although both Teacher-Only and TESSY improve performance when applied to the base model, the resulting models still lag behind Qwen3-8B by 10.99%. Moreover, compared to Qwen3-8B fine-tuned with the data synthesized by TESSY, the corresponding base model versions lag further behind by 17.67%.

This gap suggests that reasoning models, having already acquired substantial knowledge through extensive post-training, provide a stronger starting point for SFT. Discarding this learned knowledge by reverting to a base model represents a larger loss than the potential degradation caused by catastrophic forgetting. Therefore, for tasks such as code

generation, we argue that reasoning model provides a more effective starting point for SFT than base model.

Another interesting finding is that TESSY still provides advantages when applied to the base model compared to Teacher-Only. On OJBench, TESSY improved performance by 0.43% over Teacher-Only (Figure 6a), with a more pronounced gain of 7.09% on LiveCodeBench-Pro (Figure 6b). This indicates that Qwen3-8B-Base inherently has its own style, which can conflict with data generated directly by GPT-OSS. Table 5 provides an example of Qwen3-8B-Base solving a code problem, showing that it generates responses according to its own formatting, like an instruct model.

### 5.6. Changes in Data Distribution

To more clearly illustrate the differences and shifts in data distributions induced by TESSY, we sampled the same 10K questions from the training data and collected the corresponding responses for each method for visualization. Each response was tokenized using Qwen3-8B and represented as a bag-of-words vector based on TF-IDF. Figure 8 shows a PCA visualization of the resulting data distributions, comparing TESSY with Teacher-Only and Student-Only. It can be seen that the distribution of TESSY is shifted towards the student model relative to the teacher distribution, a trend that is consistently observed in three different teacher models, which may help mitigate conflicts caused by distributional differences during SFT.

## 6. Conclusion

To address the challenge of further improving reasoning models through supervised fine-tuning, we propose a teacher–student cooperation data synthesis framework, in which teacher and student models alternately synthesize responses for training. By involving the student model in the data synthesis, our approach preserves essential reasoning

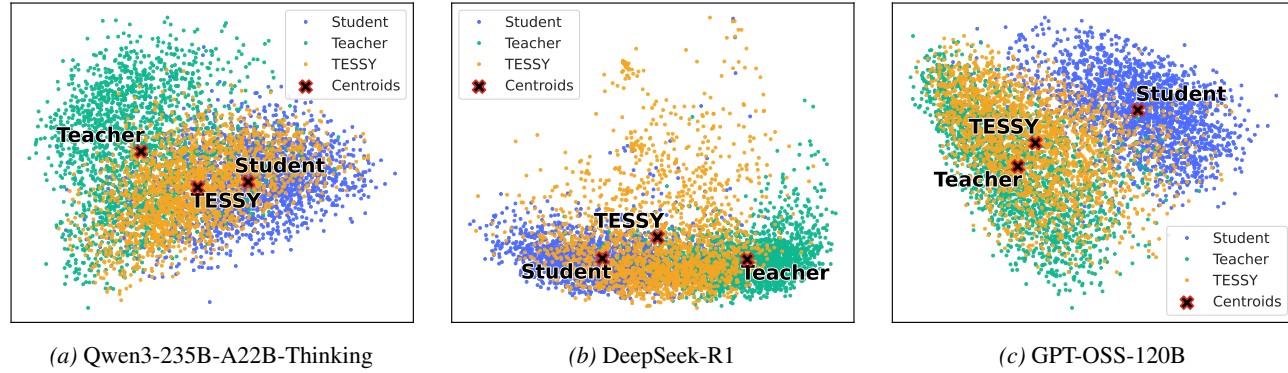

*(a)* Qwen3-235B-A22B-Thinking      *(b)* DeepSeek-R1      *(c)* GPT-OSS-120B

*Figure 8.* Visualization of the data distribution changes caused by TESSY across three teacher models.

information from the teacher while aligning the synthesized data distribution with the inherent style of the student. In the future, we will further refine the framework to enable a more accurate and efficient identification of style and capability boundaries. Due to engineering constraints, TESSY's synthesis speed remains below its theoretical limit. We will optimize the system implementation and explore acceleration techniques (Cai et al., 2024; Huang et al., 2024b; 2025). In addition, we plan to extend the proposed cooperative paradigm to a wider range of tasks and heterogeneous model settings, exploring its potential for scalable multi-model collaboration.

## Acknowledgements

This work was supported by Shanghai Artificial Intelligence Laboratory.

## Impact Statement

This paper introduces a novel data synthesis framework that alleviates the distribution mismatch in training data through a collaborative reasoning approach. From an academic perspective, our study provides new empirical evidence to understand the differences between emerging reasoning models and traditional base models. From an industrial perspective, the proposed framework offers a potentially generalizable approach to enhancing reasoning models, facilitating the development of domain-specific tasks.

In addition, we hope that the insights gained from this work can guide future research and help others avoid costly missteps. In the early stages of our study, we were very puzzled by the severe performance drops of reasoning models on code generation task after extensive SFT training. We initially misattributed the issue to factors such as insufficient response quality from the teacher model, overly simple prompts, or even potential benchmark data leakage. After an arduous investigation, we found that the primary cause was whether the training data were on-policy. Recognizing

the effort involved, we hope this work can provide not only a final solution but also insights that guide and inspire future research on reasoning models, while helping others avoid repeating the unproductive paths we explored.

## Limitations

Although, as shown in Figure 7, TESSY achieves higher code generation quality than Teacher-Only under the same maximum generation length of 40K tokens, Teacher-Only still slightly outperforms TESSY when the generation length is unrestricted, suggesting a higher upper bound of generation capability. In future work, further extending the capability upper bound of TESSY requires mitigating the quality degradation introduced by synthetic data. Moreover, the main claim of this work is that the distribution mismatch of training data is a primary factor affecting the training of reasoning models. While common data synthesis techniques, such as reject sampling, are not integrated into our data synthesis pipeline, this should not be interpreted as implying that data quality is unimportant. We advocate that, once distribution mismatches are addressed, incorporating additional techniques to continuously improve data quality can further enhance model training.

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

# A. More Comparisons

## A.1. Evaluation Beyond the Qwen3 Family

In Table 2, we evaluated the performance of TESSY when using DeepSeek-R1-Distill-Llama-8B [2] as the student model (with GPT-OSS-120B as the teacher and trained on 40K samples). The results show that TESSY consistently improves the reasoning capability of the student model, whereas self-distillation still leads to severe catastrophic forgetting.

In particular, although the Teacher-Only method performed significantly worse than TESSY, unlike the results observed in Qwen3-8B in Table 1, it improved code reasoning performance in DeepSeek-R1-Distill-Llama-8B on several benchmarks. This discrepancy may be attributed to the fact that Qwen3-8B is a more thoroughly post-trained reasoning model. As a result, introducing distributionally mismatched SFT data to a relatively well-trained model leads to more pronounced catastrophic forgetting.

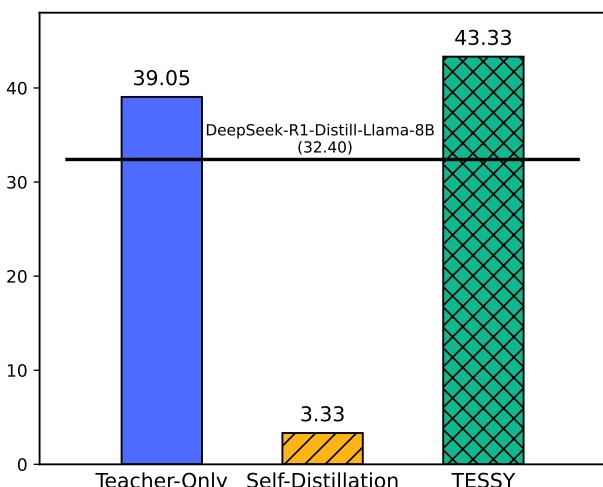

*Figure 9.* In-domain evaluation on the math reasoning task (AIME2025), where GPT-OSS-120B and DeepSeek-R1-Distill-Llama-8B were used as the teacher and student models, respectively.

## A.2. Evaluation on Math Reasoning Tasks

In Table 1, we reported the out-of-domain performance on mathematical reasoning tasks when using pure code data as the training set. To further validate the effectiveness of TESSY on other reasoning tasks, we constructed an SFT dataset using open-source math questions in this section for evaluation. However, we found that Qwen3 family models are already sufficiently trained on these datasets, such that Qwen3-8B even outperforms the teacher model GPT-OSS-120B on the training set. To provide a more faithful

evaluation, we therefore used DeepSeek-R1-Distill-Llama-8B instead of Qwen3-8B for comparison.

We randomly sampled 12K questions from DeepMath (He et al., 2025) for data synthesis, and reported the results in Figure 9. For both TESSY and the baselines, we applied rejection sampling to filter the data and retained only samples with correct answers. Consistent with previous findings, on AIME2025, TESSY outperforms Teacher-Only by up to 4.28%, while the Self-Distillation method leads to severe degradation in reasoning performance. We have published the training data synthesized by TESSY [3].

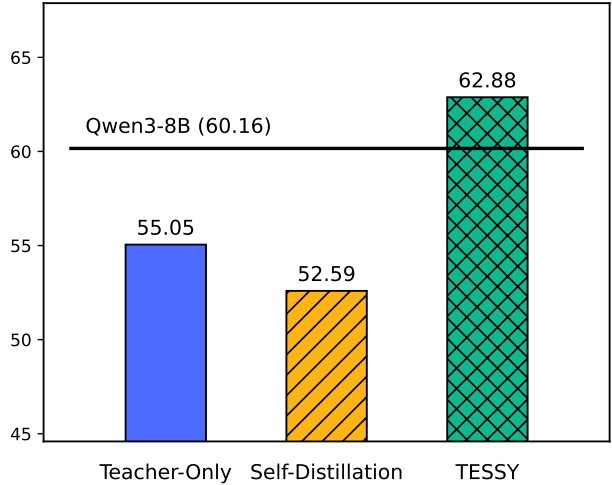

*Figure 10.* In-domain evaluation on the science reasoning task (GPQA-Diamond), where GPT-OSS-120B and Qwen3-8B were used as the teacher and student models, respectively.

## A.3. Evaluation on Science Reasoning Task

We further conducted in-domain evaluation of TESSY on GPQA-Diamond. We randomly selected 3K questions from SuperGPQA (Team et al., 2025) in Physics, Chemistry, and Biology to construct the SFT dataset. For both TESSY and the baselines, we applied rejection sampling to filter the data and retained only samples with correct answers. As shown in Figure 10, TESSY improves the performance of Qwen3-8B by 2.72%, whereas Teacher-Only leads to a 5.11% performance degradation. We have published the training data synthesized by TESSY [4].

## A.4. Comparison of Teacher Output Style Control

In Figure 11, we evaluated a baseline termed Teacher-Restyle, which conditions the teacher model with a prompt that specifies the desired style of the final answer of Qwen3-

---

[2] https://huggingface.co/deepseek-ai/DeepSeek-R1-Distill-Llama-8B

[3] https://huggingface.co/datasets/CoopReason/TESSY-Math-12K

[4] https://huggingface.co/datasets/CoopReason/TESSY-SuperGPQA-3K

*Table 2.* Performance comparison using the DeepSeek-R1-Distill-Llama-8B as the student model.

|  | LCB-V5 | LCB-V6 | LCB-Pro | OJBench |
|---|---|---|---|---|
| DeepSeek-R1-Distill-Llama-8B | 34.73 | 33.71 | 17.56 | 5.82 |
| + Self-Distillation | 19.76 | 24.00 | 8.07 | 5.82 |
| + Teacher-Only | 38.32 | 36.57 | 24.22 | 11.42 |
| + TESSY | **52.10** | **41.71** | **26.20** | **12.50** |

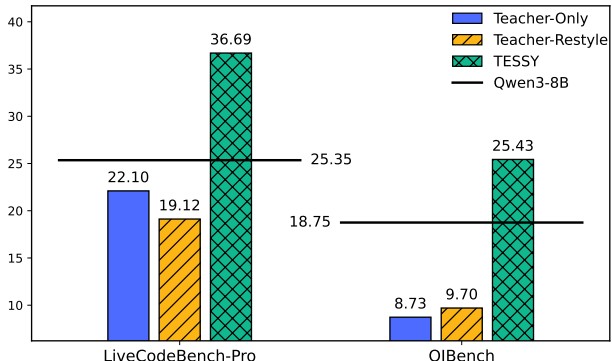

*Figure 11.* Comparison of Teacher-Restyle baseline, where we prompt teacher model to generate response as the student style.

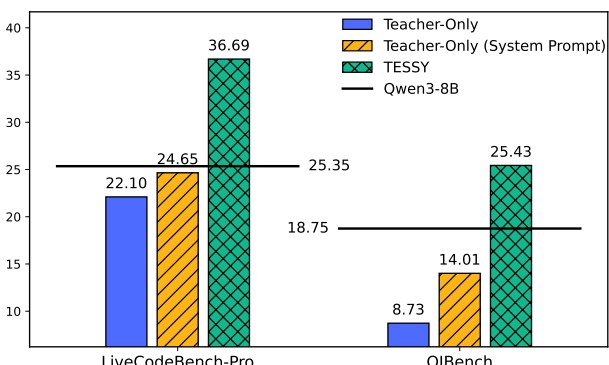

*Figure 13.* Performance comparison in TESSY with training on Teacher-Only data isolated using an independent system prompt.

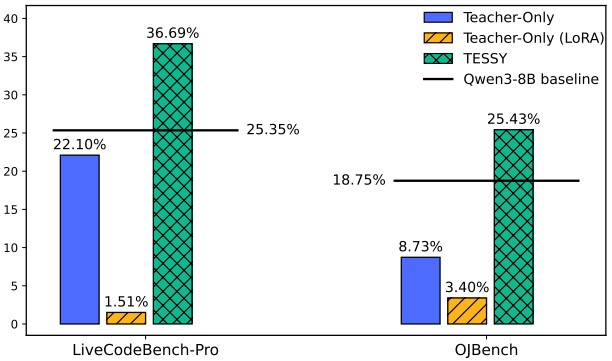

*Figure 12.* Comparison of TESSY and Teacher-Only methods trained with LoRA and full parameters.

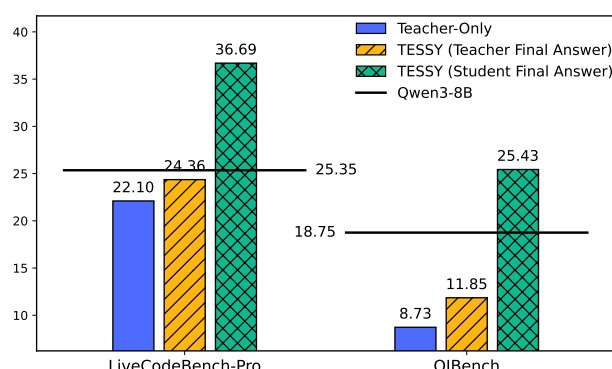

*Figure 14.* Performance comparison in TESSY with final answers generated by the teacher or the student model.

8B. Specifically, we provided an exemplar response generated by Qwen3-8B to instruct the teacher to produce reasoning traces aligned with this target output style. However, although human observation suggests that the final answers generated by the teacher model exhibit a style highly similar to that of the student model, Teacher-Restyle still leads to severe catastrophic forgetting in Qwen3-8B. This may be because the long-form thinking part is difficult to control in style through simple prompting.

## A.5. Performance of LoRA

LoRA is typically considered to mitigate catastrophic forgetting during training by freezing most of the model parameters. However, the experimental results presented in Figure 12 suggest that this may not always hold. When trained with a rank of 8 and alpha of 16 after 9 epochs, LoRA exhibited substantial performance drops on both LiveCodeBench-Pro and OJBench, even performing significantly worse than the fully fine-tuned model. These results suggest that, in reasoning models, even small parameter updates may disproportionately affect performance. Moreover, adapting to distributional differences in the data may require substantial parameter updates, which LoRA's limited update capacity may not provide.

*Table 3.* Comparison of training performance using TESSY synthetic data with boundary predictors trained on different dataset sizes. Different from Table 1 (80K samples), we used 40K synthesized samples here to reduce computational cost for comparison.

|  | LCB-V5 | LCB-V6 | LCB-Pro | OJBench |
|---|---|---|---|---|
| Qwen3-8B | 55.09 | 49.58 | 25.35 | 18.75 |
| + TESSY (Predictors trained on 500 samples) | 59.58 | 54.00 | 35.13 | 23.98 |
| + TESSY (Predictors trained on 100K samples) | **60.68** | **55.15** | **35.84** | **26.51** |

### A.6. Comparison of System Prompt Isolation

Using an independent system prompt to distinguish newly introduced SFT data from training data that the model might already have seen was a common engineering practice that helped mitigate distribution conflicts. In Figure 13, we evaluated the effect of adding an independent system prompt when training with Teacher-Only data and during inference. The results showed that with system prompt isolation, the performance of OJBench and LiveCodeBench-Pro improved by 2.55% and 5.28%, respectively, compared to the setting without system prompt. However, although system prompt isolation alleviated data distribution conflicts to some extent, it could not fully compensate for the performance degradation caused by training on stylistically different Teacher-Only data. Compared to the Qwen3-8B baseline, the performance still dropped by 0.70% on OJBench and 4.74% on LiveCodeBench-Pro, not to mention the still substantial gap compared with TESSY.

### A.7. Model Selection for Final Answer Generation

To better maintain consistency with the style of the student, TESSY uses the student model to generate the final answer instead of the teacher model. To evaluate the effectiveness of this strategy, we conducted experiments as shown in Figure 14. We observed that although the teacher model has stronger code generation ability, using it to generate the final answer led to performance drops of 12.33% and 13.58%, respectively. This indicates that avoiding style conflicts in SFT data is more important than achieving marginal improvements in data quality. Meanwhile, compared to Teacher-Only, TESSY using the teacher model to generate the final answer still achieved gains of 2.26% and 3.12% on LiveCodeBench-Pro and OJBench, respectively, further confirming the effectiveness of the alternating generation.

### A.8. Impact of Predictor Training Size

In the main experiments, we used the teacher model to automatically construct 100K training samples for both the teacher and student boundary predictors. It is worth noting that this scale is over-provisioned to better distill the teacher model's capability. In practice, boundary prediction is not a complex task, and a relatively small amount of data is sufficient for Qwen3-0.6B-Base to achieve strong

performance.

To verify the impact of training data size for the predictor on the TESSY framework, in Table 3, we compared the main setting using 100K training samples with a smaller setting using only 500 samples for training the predictors. It can be observed that, while 100K training samples allow boundary predictors to better support TESSY in synthesizing effective training data, even with only 500 samples, TESSY still functions properly, avoids catastrophic forgetting, and improves the performance of Qwen3-8B. In addition, the training data for boundary predictors is constructed fully automatically, requiring no additional human effort, and the computational cost of scaling the dataset is negligible. Therefore, in practice, we recommend using the over-provisioned training data as adopted in the main experiments of this work.

Another finding is that TESSY is already effective with only 40K training samples. In the main experiments, we used up to 80K samples and trained for 9 epochs to avoid the possibility that baselines could mitigate distribution mismatch issues under increased computational scaling. However, as shown in Table 3, we find that even when baselines are trained with twice the amount of data used by TESSY, they still fail to outperform TESSY. The effect of the number of training epochs can be found in Section B.1.

## B. More Analysis

### B.1. Performance Over Epochs

In Figure 15, we present the changes in the code generation performance of Qwen3-8B trained with TESSY synthesized data at different epochs. During the initial stages of training, the model's code generation ability temporarily decreased, likely due to adaptation to the new data distribution. By the third epoch, the model's performance had recovered to the level observed at the start of training. In subsequent epochs, its performance on LiveCodeBench-Pro and OJBench gradually improved, reaching a peak on OJBench at the 8th epoch with a score of 26.08%, while on LiveCodeBench-Pro it continued to grow, reaching 36.69% at the 9th epoch. This experiment also demonstrates that a sufficient computational budget is necessary to further enhance the reasoning capabilities of the model.

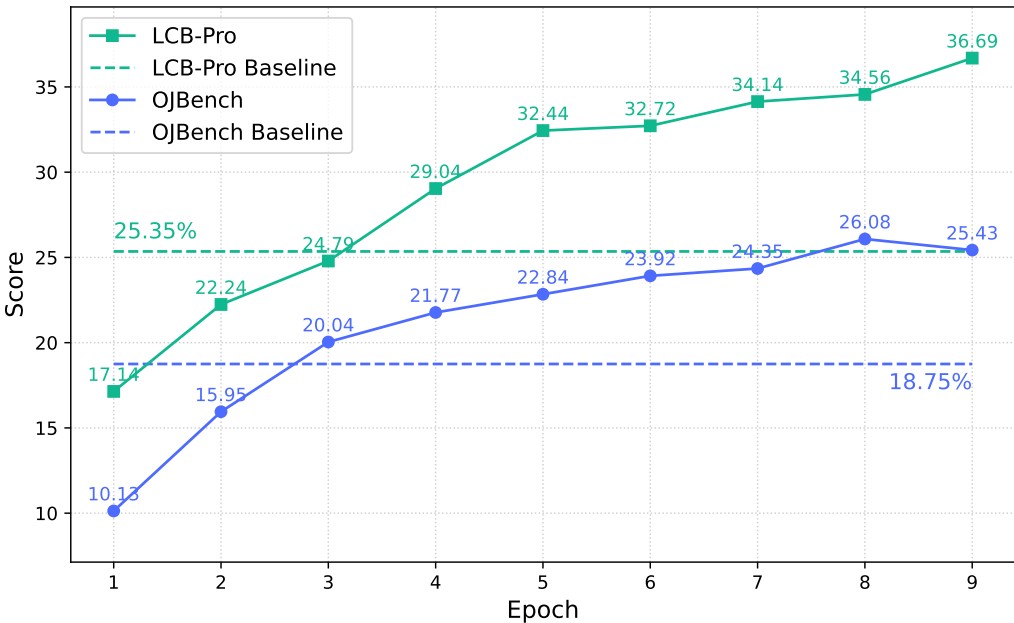

*Figure 15.* Performance of Qwen3-8B trained with TESSY (GPT-OSS-120B) across different epochs.

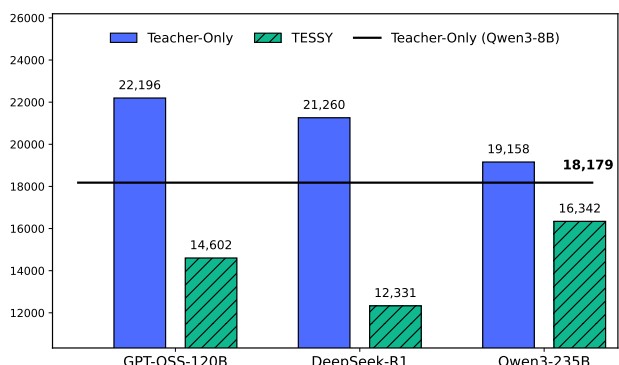

*Figure 16.* Comparison of average token counts across different synthetic data generation methods.

### B.2. Comparison of Average Token Counts

An additional benefit of TESSY is that it produces responses that are not only comparable in quality to the teacher (Figure 7) but also significantly shorter than those generated directly by the teacher. Figure 16 shows the average token counts for synthetic data generated by different methods. While the three large-scale teacher models produce responses with substantially more tokens than Qwen3-8B, responses generated under the TESSY framework are even shorter than those of Qwen3-8B. Specifically, compared to the Teacher-Only approach, TESSY reduces the average token count by 7,594, 8,938, and 2,816 when using GPT-OSS-120B, DeepSeek-R1, and Qwen3-235B-A22B-Thinking, respectively.

This phenomenon may occur because a weaker student model tends to terminate its thinking process earlier when it is exposed to teacher-enhanced reasoning trajectories. As illustrated by the examples in Table 6, in most cases, the student generates the token at the end of the thinking. This behavior not only enables more efficient training and inference, but also provides a new perspective to further improve the quality of TESSY.

### B.3. Changes in Model Output Similarity

In Figure 17, we compared the similarities of the output between different models and methods to quantify the magnitude of the differences in the data distribution and the effect introduced by TESSY. Following the same approach as in Figure 8, each sample was represented by a bag-of-words vector constructed using TF-IDF, and the average similarity between the responses under the same query was calculated to measure distributional differences between methods.

Similarly to Figure 17a and Figure 17b, it was observed that TESSY effectively increased all similarity between Qwen3-8B and all teacher models. Specifically, the similarities for Qwen3-235B-A22B-Thinking, DeepSeek-R1, and GPT-OSS-120B increased from 0.899, 0.846, and 0.849 to 0.971, 0.960, and 0.898, respectively. Moreover, compared with DeepSeek-R1 and Qwen3-235B-A22B-Thinking, the distance between GPT-OSS-120B and Qwen3-8B remains larger, which is consistent with the observations in Figure 5: training with GPT-OSS-120B under the Teacher-Only ap-

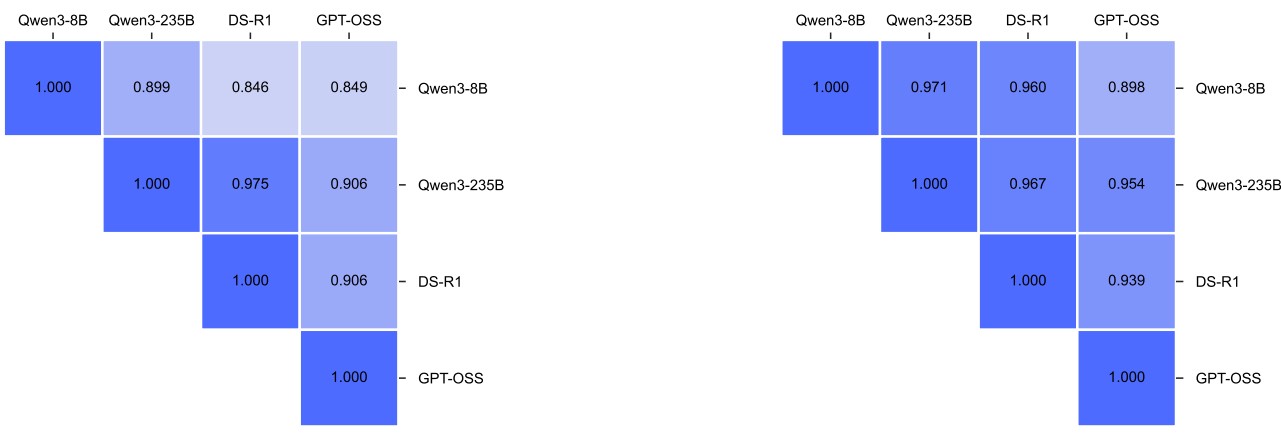

*(a)* Direct generation by the model                    *(b)* Generation under the TESSY framework

*Figure 17.* Comparison of the average output similarity between Qwen3-8B and three teacher models.

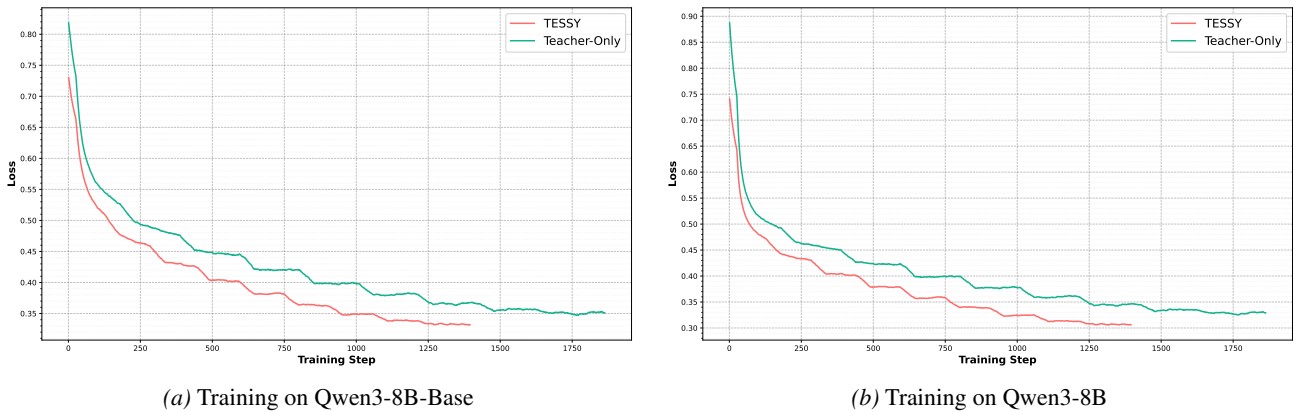

*(a)* Training on Qwen3-8B-Base                    *(b)* Training on Qwen3-8B

*Figure 18.* Comparison of training loss curves on Base and Reasoning models using SFT data synthesized by TESSY and the Teacher-Only.

proach leads to the largest performance drop among the three teacher models.

### B.4. Training Loss Curve

In Figure 18, we show the loss curves for Qwen3-8B-Base and Qwen3-8B trained with SFT data synthesized using TESSY and the Teacher-Only approach. It was observed that training with TESSY data consistently resulted in lower loss compared to Teacher-Only data. This improvement was attributed to the ability of TESSY to mitigate spurious loss caused by style tokens that did not contribute to knowledge learning. Furthermore, consistent with the discussion in Section 5.5 and the evidence presented in Figure 6, training reasoning models resulted in lower loss compared to training base models, underscoring the benefit of using reasoning models as the starting point for training. Moreover, for base models, the loss curve with TESSY remained consistently below that of Teacher-Only, suggesting that the challenge of data style conflict was a general phenomenon.

### B.5. Demonstration of Outputs

In Figure 19, we specifically compare the most frequently used words in the outputs of different models. It can be observed that certain typical high-frequency words of Qwen3-8B, such as "example" and "how", appear with much lower frequency in the outputs of other models. In Table 6, we further illustrate the differences between GPT-OSS-120B, Qwen3-8B, and the thinking content generated using TESSY. Table 7 presents the final answers generated by GPT-OSS-120B and Qwen3-8B, clearly highlighting the stylistic differences across the outputs of different models.

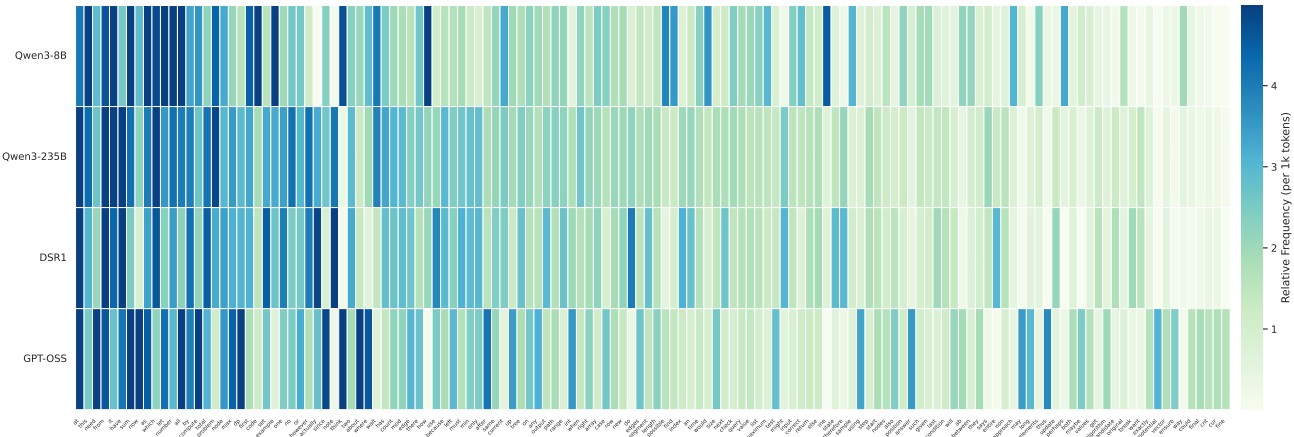

*Figure 19.* Comparison of frequently used words across models. Darker colors indicate higher word frequencies in the outputs of the corresponding models.

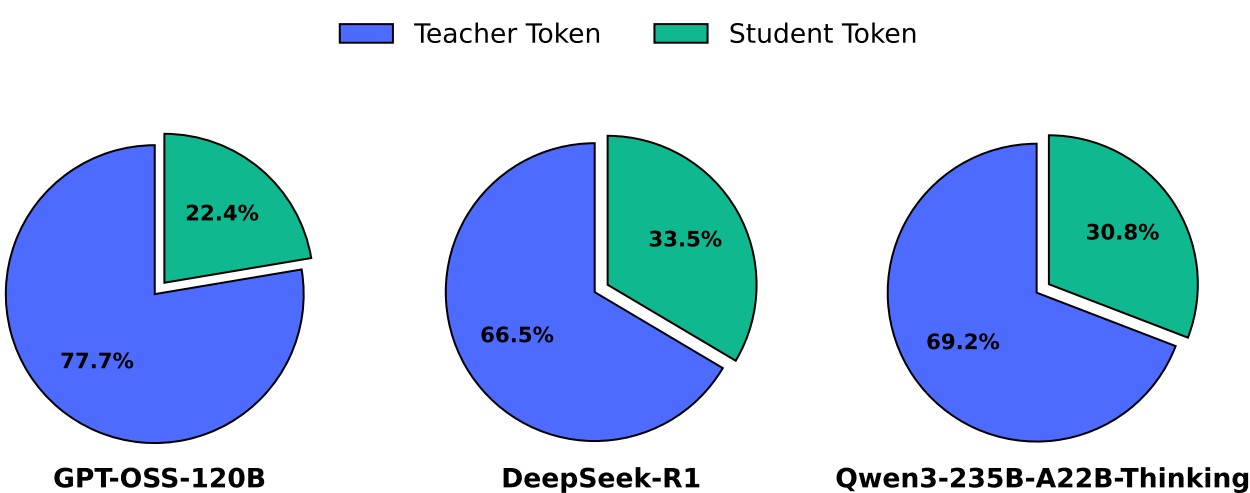

*Figure 20.* Average ratio of teacher-generated and student-generated tokens under the TESSY framework.

*Table 4.* Prompt used to annotate the training data of boundary predictor.

| Prompt |
| --- |

You are a text analysis expert.
Task: Extract all spans of text that are transitional, filler, or tone-setting phrases.
What to extract:
- Include phrases or sentences that:
    - Express hesitation, tone, or attitude (e.g., "well", "okay", "so", "let's see", "I think")
    - Indicate transition or setup (e.g., "to begin with", "in this case", "for example", "but if")
    - Serve as narration or connection, not analysis
- Do not include:
    - Actual reasoning, deduction, or explanation
    - Code or formula descriptions
    - Problem-solving steps

Output format (STRICT JSON):
- Return a JSON array of strings, e.g.: `["", "", ...]`
- Rules:
    1. Each span must be copied verbatim from the original text.
    2. Preserve order of appearance.
    3. If there are none, return an empty list: `[]`
    4. Output only the JSON array — no explanation or extra text.

`<input_text> {think_text} </input_text>`

*Table 5.* Example output of Qwen3-8B-Base

| Problem Description |
| --- |

Given a transcript of a Fizz Buzz game (where numbers divisible by $a$ are replaced with "Fizz", by $b$ with "Buzz", and by both with "FizzBuzz"), find any valid values of $a$ and $b$ that could generate the transcript.
**Input:** Two integers $c$ and $d$ ($1 \leq c \leq d \leq 10^5$) indicating the range, followed by $d - c + 1$ tokens (integers or "Fizz"/"Buzz"/"FizzBuzz").
**Output:** Two positive integers $a$ and $b$ ($1 \leq a, b \leq 10^6$) consistent with the transcript. If multiple solutions exist, output any one.

| Response of Qwen3-8B-Base |
| --- |

To solve this problem, we need to determine the values of $a$ and $b$ that could have generated the given transcript. The key insight is that the transcript consists of numbers and strings ("Fizz", "Buzz", "FizzBuzz") that correspond to multiples of $a$ and $b$ ...
To find $a$ and $b$, we can: ...
- Identify the positions of "Fizz" and "Buzz" in the transcript.
- ...

Approach
- Identify the positions of "Fizz" and "Buzz": ...

Solution Code
```
#include <iostream>
```
`...`
Explanation
- Input Reading:
    - The program reads the range $[c, d]$ and the transcript.
- Position Identification:
    - The program identifies the positions of "Fizz" and "Buzz" in the transcript.
- Validation Loop:
    - The program iterates over possible values of $a$ and $b$ and checks if they satisfy the conditions for the transcript.
- Output:
    - The first valid pair $(a, b)$ is output, and the program terminates.

This approach ensures that we find a valid pair $(a, b)$ that could have generated the given transcript.

*Table 6.* Comparison of thinking content generated by the teacher, student, and TESSY for the same programming problem, with TESSY portions generated by the student highlighted in **purple**.

---

**Problem Description**

---

Given a transcript of a Fizz Buzz game (where numbers divisible by $a$ are replaced with "Fizz", by $b$ with "Buzz", and by both with "FizzBuzz"), find any valid values of $a$ and $b$ that could generate the transcript.
**Input:** Two integers $c$ and $d$ ($1 \leq c \leq d \leq 10^5$) indicating the range, followed by $d - c + 1$ tokens (integers or "Fizz"/"Buzz"/"FizzBuzz").
**Output:** Two positive integers $a$ and $b$ ($1 \leq a, b \leq 10^6$) consistent with the transcript. If multiple solutions exist, output any one.

---

**Thinking of Student Model (Qwen3-8B)**

---

`<think>` Okay, let's see. I need to solve this Fizz Buzz reverse problem ...
First, I need to understand what the transcript implies about a and b ...
Let me think about how to model this ...
But wait, the transcript could have numbers that are not divisible by a or b. So for each entry in the transcript, we can derive constraints on a and b.
The challenge is to find a and b that satisfy all these constraints.
Let me think about how to approach this ...
But how do I find a and b that satisfy all these constraints?
...
Thus, the code is implemented as described. `</think>`

---

**Thinking of Teacher Model (GPT-OSS-120B)**

---

`<think>` We need to solve the reverse FizzBuzz problem: given a transcript from c to d inclusive, each entry is either a number (as integer string) or one of the words "Fizz", "Buzz", "FizzBuzz". The transcript is guaranteed to be consistent with some a,b ($1 \leq$ a,b $\leq$ 1e6). Need to output any such a,b.
Constraints: c,d up to 1e5 length (max length of transcript is up to 1e5). a,b up to 1e6.
Goal: find any pair (a,b) that matches.
We need to parse transcript: for each integer i from c to d, we have output string s_i. If s_i is a number, it must be exactly the integer i (as decimal). If s_i is "Fizz", then i is multiple of a but not b. If "Buzz": multiple of b but not a. If "FizzBuzz": multiple of both a and b (i.e., multiple of lcm(a,b)) ...
We need to find any a,b. This is like solving constraints.
Observation: The numbers in transcript that are numeric give us direct constraints: ...
. . .
Let's write code.`</think>`

---

**Thinking of TESSY (Teacher: GPT-OSS-120B, Student: Qwen3-8B)**

---

`<think>` Okay, I need to solve the problem: given a transcript of FizzBuzz from integer $c$ to $d$ inclusive, ...
Edge Cases: If $F$ is empty (no "Fizz" or "FizzBuzz" tokens). ... So if $d \leq 10^6$, we can set $a = d + 1$ (if $\leq 10^6$). But that's possible.
But how to find $a$ and $b$? Let me think of the approach:
- Parse tokens into list tokens$[0..\text{len} - 1]$ corresponding to $n = c + i$.
That's part of NB_for_b (since set_fizz $\in$ NB_for_b). So for $b$, we need $b$ not dividing any $n$ in NB_for_b ...
But since $a$ divides all fizzbuzz numbers (by dividing $G_f$) and $b$ also divides all fizzbuzz numbers (by dividing $G_b$), they both will divide them. So the independent selection works ...
Now, code:
...
Edge Cases: Case where transcript includes only Fizz and FizzBuzz, no Buzz, no numbers. Example: $c = 1$, $d = 5$ tokens: "Fizz 2 3"? ... Let's think of a case where all tokens are "Fizz" or "FizzBuzz"...That would require that $a = 1$ (so every number is multiple of $a$) and $b$ is something that divides some numbers (Buzz) but we don't have any Buzz tokens. So that's impossible unless $b$ is also 1? Not sure. But regardless, ...
...
Thus, the code should handle all cases.
Now, let's code it. `</think>`

---

*Table 7.* Comparison of final answer content generated by the teacher and student models for the same programming problem

---

**Problem Description**

---

Given a transcript of a Fizz Buzz game (where numbers divisible by $a$ are replaced with "Fizz", by $b$ with "Buzz", and by both with "FizzBuzz"), find any valid values of $a$ and $b$ that could generate the transcript.
**Input:** Two integers $c$ and $d$ ($1 \leq c \leq d \leq 10^5$) indicating the range, followed by $d - c + 1$ tokens (integers or "Fizz"/"Buzz"/"FizzBuzz").
**Output:** Two positive integers $a$ and $b$ ($1 \leq a, b \leq 10^6$) consistent with the transcript. If multiple solutions exist, output any one.

---

**Final Answer of Student Model (Qwen3-8B)**

---

**To solve the Fizz Buzz reverse problem**, we need to determine possible values of integers $a$ and $b$ that could generate a given transcript of the game. The transcript consists of numbers and the words "Fizz", "Buzz", and "FizzBuzz" corresponding to multiples of $a$, $b$, and both $a$ and $b$ respectively.

**Approach**
- Parse Input: Read the range of the transcript and the transcript itself.
- Classify Entries: Identify which entries are "Fizz", "Buzz", "FizzBuzz", or numbers. This helps in determining constraints on $a$ and $b$.
- Determine Constraints: For entries that are "Fizz" or "FizzBuzz", $a$ must divide the corresponding number...
- Generate Candidates: For "Fizz" and "FizzBuzz" entries, compute the GCD of all such numbers and generate all its divisors as potential values for $a$ ...
- Validate Candidates: Check all combinations of $a$ and $b$ to find a valid pair that satisfies all constraints from the transcript.

**Solution Code**
```
#include <bits/stdc++.h>
using namespace std;
...
```

**Explanation**
- Input Parsing: The input is read and stored in a vector of strings representing the transcript.
- Classifying Entries: We classify each entry to determine which numbers are divisible by $a$, $b$, or both.
- Generating Candidates: Using the GCD of relevant numbers, we generate potential values for $a$ and $b$. If no such entries exist, we find the smallest number that does not divide any number in the transcript.
- Validation: For each candidate pair $(a, b)$, we check if it satisfies all constraints from the transcript ...

This approach ensures that we efficiently narrow down the possible values of $a$ and $b$ and validate them against the transcript constraints.

---

**Final Answer of Teacher Model (GPT-OSS-120B)**

---

**Solution Explanation**
For every integer `i`, ...
1. **Reformulating the conditions**
   ```
   S_Fizz = { i | type(i) = Fizz }
   ```
   ...
2. **Determining `a`**
   If A is empty, `a` never has to divide any number, we may choose any value that ...
3. **Algorithm**
   ```
   read c , d
   ```
   ...
4. **Correctness Proof**
   - Lemma 1: If A is non-empty, any integer x that divides every element of A must divide `G = gcd(A)`; conversely every divisor of G divides every element of A.
     Proof: ...
   - ...
5. **Complexity Analysis**
   $N = d-c + 1 (\leq 10^5)$
   ... Both limits are easily satisfied.
6. **Reference Implementation**
   ```
   #include <bits/stdc++.h>
   using namespace std;
   ...
   ```
   The program follows exactly the algorithm proven correct above and conforms to the GNU C++17 compiler.

---

