# OpenReview forum: "How to Fine-Tune a Reasoning Model? A Teacher–Student Cooperation Framework to Synthesize Student-Consistent SFT Data"
_ICML.cc/2026/Conference — ICML 2026 regular_

### Official Review · Reviewer_dhnr · 2026-03-11

**Soundness:** 3
**Presentation:** 3
**Significance:** 3
**Originality:** 3
**Overall Recommendation:** 5
**Confidence:** 4

**Summary:**

This paper studies the field of distillation from a teacher (equivalently synthetic data generation for the purpose of training a student model). The specific problem studied by the paper is a _distribution mismatch_ between the teacher and the student. Specifically, the teacher model's reasoning traces are off-policy for the student. Their solution is (roughly speaking) to decouple the semantic structure of the reasoning traces from the stylistic presentation of the reasoning traces. They show that this performs significantly better than directly distilling from the teacher.

**Compliance With Llm Reviewing Policy:**

Affirmed.

**Final Justification:**

I asked for two things in the rebuttal:
1. A comparison to stronger baselines.
2. Results on more domains.

The authors provided both and I raised my evaluation as my main concerns were answered. I think this paper tackles a meaningful and important problem (off-policyness of distillation) with a relatively simple solution that also seems to work well, seems scalable, and is general. Another strength of this paper is that it works on a very strong post-trained model (e.g. Qwen3). This is not easy to do (AFAIK, previous attempts to distill to a highly-trained model like Qwen3 have required much larger scale).

**Key Questions For Authors:**

1. Can you compare to a self-distillation / prompted baseline where the student rewrites the reasoning trace given the reasoning trace of the teacher?
2. Can you compare to a self-distillation / prompted baseline where the teacher rewrites the reasoning trace given the style of the student? This is not as important as (1) and can be skipped.
3. Does this approach work on things besides code generation?

If the boundary predictor approach beats a prompted style transfer baseline / self-distillation baseline, I will raise my score to a Weak Accept. If promise can be demonstrated on things besides code gen (e.g. mathematics, or any other reasoning task), I will raise my score to an Accept.

**Limitations:**

Yes

**Strengths And Weaknesses:**

I quite like this paper — the motivation is clear and the approach is relative simple (a strength). If I understand correctly, the idea is to essentially do a sort of style transfer to turn off-policy data into on-policy data, and the obstacle cleared by the authors is "how do we determine what parts of the teacher's reasoning traces are stylistic vs semantic?"

I think the solution is clever and makes sense. The solution is to train a model that predicts which tokens are stylistic tokens vs semantic tokens and alternate between the teacher and the student. I did think the presentation of the method could be improved with some concrete examples, it was not obvious that the boundary predictor is _token level_ but the teacher and student alternate based on _spans_.

The primary weakness of the paper is that the baselines are IMO, a bit gimped, and there is no ablation of the primary design choices (as I understand them). There should probably be an ablation or baseline that is prompted. For example, what if we just ask the teacher (or student) to rewrite the reasoning traces in the style of the student? This is even mentioned in the related works (L408-413, right-hand column).

Btw, I appreciated the quality and thematic consistency of the figures (the consistent coloring was very helpful!)

---

> ### Author Rebuttal · Authors · 2026-03-31
>
> We sincerely thank the reviewer for the positive assessment of our work. We are especially encouraged by your recognition of our motivation, methodology, and the presentation of our paper.
>
> In particular, we appreciate that you have clearly identified two key concerns that are critical for improving your score. These points are very helpful for guiding our discussion and will enable us to further strengthen and refine the paper. Our responses are as follows.
>
> ## Q1 of Reviewer dhnr. Compare to a self-distillation / prompted baseline where the student rewrites the reasoning trace given the reasoning trace of the teacher.
>
> In fact, the submitted version of our paper **already includes a self-distillation baseline in Table 1**, which we **denote as Teacher-Reference**, i.e., the teacher provides a reference answer and the student rewrites the reasoning trace.
>
> Our experimental results show that self-distillation can even harm reasoning performance, leading to consistent degradation on both code and mathematical reasoning tasks. We hypothesize that, when a reference answer is provided, the student tends to introduce unreasonable shortcuts while rewriting the reasoning traces. As shown in Figure 11, we measure the average generation length on the training set: the Qwen3-8B baseline produces 18,179 tokens on average and TESSY reduces this to 14,602 tokens, while self-distillation further shortens it to 13,389 tokens.
>
> Our hypothesis is further supported by recent work from Microsoft Research (https://arxiv.org/pdf/2603.24472), which finds that providing a teacher’s reference answer encourages the student to imitate a confident reasoning style that presupposes information. This effect is particularly detrimental in complex tasks such as mathematics and coding. Their experiments show that self-distillation can **lead to a performance drop of up to 40% on AIME24**, aligning with our observations.
>
> We thank the reviewer for noting that our comparison to self-distillation was not clearly recognized, highlighting the need for clearer presentation. For consistency, we named all baselines in the form Teacher-XXX, but this may have caused some confusion. Additionally, there is a typo on Line 234: Teacher-Reference was incorrectly cited (the correct reference is "Self-distillation bridges distribution gap in language model fine-tuning"). We will clarify these in the next revision.
>
> ## Q2 of Reviewer dhnr. Compare to a self-distillation / prompted baseline  where the teacher rewrites the reasoning trace given the style of the student
>
> We thank the reviewer for the suggestion. In our experiments, we specify the style of the final answer of Qwen3-8B in the prompt, provide an example of a Qwen3-8B generated final answer, and instruct the teacher model to generate reasoning traces consistent with the Qwen3-8B style. We named this method as Teacher-Rewrite and present the experimental results as below.
>
> |                                              |   LCB-V5   |   LCB-V6   |  LCB-Pro   |  OJBench   |
> |----------------------------------------------|:----------:|:----------:|:----------:|:----------:|
> | Qwen3-8B                                     |   55.09    |   49.58    |   25.35    |   18.75    |
> | + Teacher-Only                               |   41.32    |   36.57    |   22.10    |    8.73    |
> | + Teacher-Reference (Self-Distillation)        |   51.50    |   43.43    |   19.26    |   14.22    |
> | + Teacher-Rewrite (Given the style of student) |   35.93    |   33.71    |   19.12    |    9.70    |
> | + TESSY                                        | **62.87**  | **55.43**  | **36.69**  | **25.43**  |
>
> Although we can observe that the teacher generates final answers structurally consistent with Qwen3-8B, and they appear indistinguishable to human inspection, the style of the thinking portion remains difficult to control, and finer stylistic details are hard to enforce. As a result, the Teacher-Rewrite setting leads to a substantial drop in reasoning performance.
>
> ## Q3 of Reviewer dhnr. Working on mathematical and scientific reasoning.
>
> Due to space constraints, we present our experimental results on Math Reasoning and Scientific Reasoning in ``Q1 of Reviewer nTeU`` and ``Q1 of Reviewer QVvU``, respectively.
>
> ## We Welcome Your Further Comments
>
> We sincerely thank you again for your constructive comments. We note that you rated each of Soundness, Presentation, Significance, and Originality as *Good*, while keeping a cautious initial Overall Recommendation score. If any of our responses still fail to address your concerns, please kindly let us know. We will do our best to address them within the remaining rebuttal period.
> We hope our method will be widely adopted as an improved alternative to self-distillation, so we consider all of your feedback to be invaluable.

---

> > ### Author Rebuttal · Reviewer_dhnr · 2026-03-31
> >
> > My main question was regarding the rewrite baseline and results on other domains. The authors have provided a comparison to the requested baseline and results on other domains in their responses to other reviewers. I have raised my rating to a clear accept, as I have a positive view of the paper and no other remaining questions.

---

> > > ### Author Response · Authors · 2026-04-02
> > >
> > > Thank you for your timely response and for recommending our paper be accepted. We will carefully incorporate your suggestions into the final version.
> > >
> > > Thank you again for your support！

---

### Official Review · Reviewer_nTeU · 2026-03-11

**Soundness:** 3
**Presentation:** 3
**Significance:** 3
**Originality:** 3
**Overall Recommendation:** 4
**Confidence:** 4

**Summary:**

This paper studies the question: can we fine-tune a reasoning model using synthetic SFT data from a stronger teacher, while maintaining consistency with the distribution of the student model. First, the paper provides observation that for reasoning models, teacher-only synthetic data can hurt model performance. To solve the problem, authors proposed a Teacher–Student Cooperation Data Synthesis framework (TESSY), which alternately uses student and teacher models to generate stylistic content and reasoning content, respectively. TESSY uses a generate-then-rollback strategy, allowing fine-grained control over the responsibilities of the teacher and student models. Results on code and math benchmarks show that using GPT-OSS120B as the teacher to train the stylistically distinct student Qwen3-8B, results in performance drops of up to 3.25% and 10.02% in LiveCodeBench-Pro. In contrast, TESSY increases the performance of Qwen3-8B by 11.25% and 6.68% on the same benchmarks

**Compliance With Llm Reviewing Policy:**

Affirmed.

**Final Justification:**

My concerns are resolved. So I recommend acceptance for this paper

**Key Questions For Authors:**

1. (line 184-190) How reliable are the capability and style boundary annotations used to train the predictors? My concern is that If these annotations contain noisy information, the method may become less stable than reported.

2. Does TESSY perform good on non-code reasoning tasks when those are used as the main training target rather than only out-of-domain evaluation?

**Limitations:**

Yes

**Strengths And Weaknesses:**

1. The paper is well motivated. Using synthetic data from a larger model is important, however, many existing work ignores that the student's own distribution many matter a lot during SFT. The paper provides a very useful praticle insights.

2. The proposed method is simple and effective. Alternating generation + rollback is easy to understand, and empirically shows better performance than prior methods.

3. Experiments are good and diverse, covering different teachers and students, and also include cross-trianing analysis. These experiments make the empirically story solid.

4. The paper evaluates model mostly on coding tasks, though some math/science numbers are helpful, but I am not fully sure whether this is a broad method for “reasoning models” or mostly a code-SFT recipe.

5. The boundary predictors are trained using 100k spans from the teacher- and student with a carefully crafted prompt. And this needs more discussion of annotation quality,  noise and errors.

---

> ### Author Rebuttal · Authors · 2026-03-31
>
> We thank the reviewer for recognizing the motivation of our work, as well as the value of our method being simple and effective, and the comprehensiveness of our experiments. We understand that the reviewer still has concerns regarding the generalization of our method, including the boundary predictor and its performance on non-code reasoning tasks. We greatly appreciate the reviewer’s valuable feedback and provide our response as follows.
>
> ## Q1 of Reviewer nTeU. Does TESSY perform good on non-code reasoning tasks.
>
>
> ### **Why we prioritized code generation task over math in the submitted version?**
>
> The primary reason we prioritized code tasks over math reasoning tasks in the submitted version of the paper, besides their popularity and broad practical applicability, is that **Qwen3 models are at risk of overfitting on math tasks**.
>
>
> |         | AIME2025 (Test set) | DeepMath-103K (Training Set) |         Big-Math-RL-Verified  (Training Set)       |      Nemotron-RL-math-OpenMathReasoning (Training Set)      |
> |--------------------------|:-------------------:|:----------------------------:|:------------------------------------:|:---------------------------------------------:|
> | Qwen3-8B                 |        69.17        |          **75.25**           |              **54.66**               |                   **16.47**                   |
> | GPT-OSS-120B             |      **80.33**      |            34.26             |                47.49                 |                     16.34                     |
>
>
> As shown in the table above, while the teacher model GPT-OSS-120B outperforms the student model Qwen3-8B on the AIME2025 test set as expected, we observe, however, that Qwen3-8B surpasses GPT-OSS-120B on several influential open-source training sets under a maximum length of 40K.
> In particular, on DeepMath, Qwen3-8B significantly surpasses GPT-OSS-120B.
> We think this indicates a risk of overfitting in Qwen3-8B, raising concerns that using these datasets to synthesize SFT data for Qwen3-8B may introduce noise that is unrelated to the objectives of our study.
>
>
> ### **We provide the experiments on math task.**
>
> However, given the short rebuttal period, to quickly address the reviewer’s concern, **we conduct experiments on math task using DeepSeek-R1-Distill-Llama-8B as the student model**.
>
> | Math    Reasoning    Task                        |  AIME24   |   AIME25   |  OlympiadBench  |
> |-----------------------------------------|:---------:|:----------:|:---------------:|
> | DeepSeek-R1-Distill-Llama-8B            |   51.98   |   32.40    |      48.71      |
> | + Teacher-Only                          |   52.31   |   39.05    |      49.31      |
> | + Teacher-Reference (Self-Distillation) |   11.56   |    3.33    |      10.58      |
> | + TESSY                                 | **57.60** | **43.33**  |    **53.45**    |
>
> We randomly sampled 12K questions from DeepMath for our experiments, with results reported in the table above. TESSY consistently improves the student model across all three math reasoning tasks, significantly outperforming the Teacher-Only approach. In contrast, the Self-Distillation method, which has been highlighted by other reviewers, introduces unreasonable shortcuts in the reasoning traces, resulting in substantial performance degradation (More discussion can be seen in ``Q2 of Reviewer dhnr``).
>
> We are still in the process of identifying suitable math training data for Qwen3-8B and conducting experiments, and we will report the performance of math-data SFT on Qwen3-8B in the next version of the paper.
>
> ### **We also add experiments on the scientific reasoning dataset GPQA-Diamond.**
>
> Due to space constraints, please refer to the response of ``Q1 of Reviewer QVvU`` for the results.
>
>
> ## Q2 of Reviewer nTeU. How reliable are the capability and style boundary
>
> We trained a predictor using only 500 samples from code tasks. In ``Q2 of Reviewer zXPC``,
> we compared it with a predictor trained on 100K samples and found that **500 samples are already sufficient to synthesize high-quality data.**
> Furthermore, the experiments on different student models and tasks reported in ``Q1 of Reviewer QVvU`` and ``Q3 of Reviewer QVvU``, as well as ``Q1 of Reviewer nTeU``, all use data synthesized by this 500-sample predictor. All results are positive, demonstrating that the predictor does not become a bottleneck for our method.
>
>
> ## Welcome Your Further Comments
> We sincerely thank you for your professional review and welcome any further questions you may have.
> Moreover, **we commit to publicly releasing all the training datasets we synthesized in our experiments**, including code, math, and scientific reasoning tasks, as well as any additional resources you may suggest.
> We hope to address your concerns and potentially further improve your assessment of our work.

---

> > ### Author Rebuttal · Reviewer_nTeU · 2026-04-04
> >
> > I decided to main my positive score

---

> > > ### Author Response · Authors · 2026-04-04
> > >
> > > We sincerely thank the reviewer for your positive recommendation of our work.
> > >
> > > If there are any additional concerns that we could address to further improve the assessment of our work,  we would be glad to address them.
> > >
> > > Thank you again for your time and feedback!

---

### Official Review · Reviewer_QVvU · 2026-03-12

**Soundness:** 3
**Presentation:** 4
**Significance:** 2
**Originality:** 3
**Overall Recommendation:** 4
**Confidence:** 3

**Summary:**

Distilling and then fine-tuning synthetic data/reasoning traces from strong models is an effective way to improve the capabilities of smaller models. However, this is not always effective, as the smaller model may have significant distributional differences from the parent model. This work proposes TESSY, a method to modify the *style* of the distilled traces post-hoc to reduce distributional drift and improve the efficacy of the teacher-generated data. The method is successful across a variety of coding benchmarks, while retaining performance on OOD test sets (not overfitting).

**Compliance With Llm Reviewing Policy:**

Affirmed.

**Final Justification:**

My original concerns were task generalization beyond code, missing baselines, and limited model diversity. The rebuttal addressed all three: TESSY works on science and math reasoning, Teacher-Reference already covers self-distillation, and DeepSeek-R1-Distill-Llama-8B results demonstrate cross-family effectiveness. The core idea is sound and well-executed, though the significance remains somewhat incremental given the method's complexity. Raising from 3 (weak reject) to 4 (weak accept).

**Key Questions For Authors:**

None

**Limitations:**

No, they did not discuss limitations and potential negative societal impact of their work. They could potentially talk about misuse potential of this or mention concrete limitations.

**Strengths And Weaknesses:**

Strengths
* This paper is written well and easy to follow
* The paper is timely and the problem is well-motivated and relevant to real-life workflows today.
* The idea is interesting - converting the style of the traces - and executed well
* The proposed method improves performance on Qwen3-8B on a variety of coding benchmarks while retaining OOD performance on different math/reasoning tasks
* The ablations are helpful and answer several interesting questions (e.g., training on SFT vs base model). The visualizations shown in Figure 8 also nicely show the distribution shift in TF-IDF.

Weaknesses
* The primary tasks are code which may have a stronger split between style and capability (ie, examples that naturally fit in the format shown in Figure 1) . The results show that OOD performance remains around the same level while coding capabilities increase, but it would be helpful to show this / other reasoning tasks as the target metric to improve
* There are several missing related work/baselines which seem important to compare against and establish novelty. 1) Speculative Knowledge Distillation (https://arxiv.org/abs/2410.11325v1) and GRAPE (https://arxiv.org/abs/2505.20380). Self-distillation also seems to be a natural baseline as it explicitly tries to solve the same problem of making student-aligned data.
* It would be more convincing (but not necessary) if the the model diversity is expanded beyond 1) Qwen family and 2) 8B size

---

> ### Author Rebuttal · Authors · 2026-03-31
>
> We sincerely thank you for your review and the time you have dedicated. Your main concerns focus on the generalization of our method to other reasoning tasks and the potential lack of certain baselines. We provide our responses below.
>
> ## Q1 of Reviewer QVvU. Show other reasoning tasks.
>
> Due to space constraints, we present here only the results on the Scientific Reasoning Task. Results for Math Reasoning can be found in ``Q1 of Reviewer nTeU``.
>
> | Scientific Reasoning Task                    | GPQA-Diamond |
> |----------------------------------------------|:------------:|
> | Qwen3-8B                                     |    60.16     |
> | + Teacher-Only                               |    55.05     |
> | + Teacher-Reference (Self-Distillation)      |    52.59     |
> | + TESSY                                      |  **62.88**   |
>
> For the GPQA-Diamond dataset used in our paper,
> we selected a subset of SuperGPQA consisting of questions labeled with Physics, Chemistry, and Biology,
> and randomly sampled 3K examples for our experiments.
> As with the code and math tasks, TESSY improves the model’s scientific reasoning ability, significantly outperforming the Teacher-Only approach, while the Self-Distillation method harms reasoning performance.
>
>
> ## Q2 of Reviewer QVvU. Related baselines.
>
> ### **Speculative Knowledge Distillation**
> Methods of this type typically **require the teacher and student models to belong to the same family and use the same tokenizer.**
>
> **Our method supports teachers and students from different model families (e.g., GPT-OSS-120B and Qwen3-8B), which better reflects practical scenarios.**
>
> Moreover, our work aims to synthesize SFT data that can be openly shared, which is fundamentally different from knowledge distillation methods. We will further cite and discuss these methods in the next version of our paper.
>
> ### **GRAPE**
>
> | Baseline inspired by GRAPE | LCB-V5 | LCB-V6 | LCB-Pro | OJBench |
> |------------|:------:|:------:|:-------:|:-------:|
> | Qwen3-8B   | 55.09  | 49.58  | 25.35   | 18.75   |
> | + Teacher-Mix | 45.51  | 40.57  | 21.95   | 11.85   |
> | + Teacher-Mix (Smooth) | 45.51  | 38.86  | 22.52   | 14.66   |
> | + TESSY    | **62.87**  | **55.43**  | **36.69**   | **25.43**   |
>
> GRAPE appears to be a data selection method rather than a data synthesis method,
> and its code is not publicly available, making short-term reproduction challenging.
> **A related baseline, Teacher-Mix, is already included in Table 1 of our submitted paper**.
> It trains the model on data generated from a mixture of teacher and student outputs.
> Inspired by the curriculum learning strategy in GRAPE, we further improved the Teacher-Mix method by starting with a smaller proportion of teacher data in the initial epochs and gradually increasing it. Specifically, teacher samples are drawn at a ratio of $epoch / (2×epoch_total_num)$ . The results, shown in the table above, indicate that TESSY remains the method with a significant lead.
>
> ### **Self-distillation**
>
> **In Table 1 of our submitted paper, we have already compared with self-distillation**, which we refer to as Teacher-Reference. As discussed in the response of ``Q1 of Reviewer dhnr``, applying self-distillation to SFT reasoning models can degrade model performance.
>
>
> ## Q3 of Reviewer QVvU. Expand the model diversity.
>
> ### **Beyond Qwen family**
>
>
> | Different Students                      | LCB-V5 | LCB-V6 | LCB-Pro | OJBench |
> |-----------------------------------------|:------:|:------:|:-------:|:------:|
> | DeepSeek-R1-Distill-Llama-8B            | 34.73  | 33.71  | 17.56%  |  5.82  |
> | + Teacher-Only                          | 38.32  | 36.57  | 24.22%  |  11.42 |
> | + Teacher-Reference (Self-Distillation) | 19.76  | 24.00  |  8.07%  |  5.82  |
> | + TESSY                                 | **52.10**  | **41.71**  |  **26.20**  | **12.50**  |
>
> **In Figure 5 of our paper, we have already evaluated teacher models from different families**. In the table above, we further experiment with student models from different families within our framework. It can be seen that when using DeepSeek-R1-Distill-Llama-8B as the student model, TESSY still demonstrates very strong performance.
>
> ### **Beyond 8B size**
>
> **In Figure 3 of our submitted paper, we have already compared with the MoE model Qwen3-30B-A3B**. The results demonstrate that TESSY remains effective even on 30B-scale models.
>
> ## We look forward to your feedback
>
> We sincerely thank the reviewer for your time and effort.
> We are greatly encouraged by your positive remarks that “The paper is timely” and “relevant to real-life workflows.”
> We note that you rated the paper as good for Soundness and Originality, and excellent for Presentation, while the Overall Recommendation is relatively cautious.
> We welcome and look forward to any further feedback, and we are happy to address any additional concerns to resolve all your doubts.

---

> > ### Author Rebuttal · Reviewer_QVvU · 2026-04-06
> >
> > Thanks for the thorough rebuttal, and for clarifying that Teacher-Reference serves as a self-distillation baseline. The new results on scientific reasoning, math, and DeepSeek-R1-Distill-Llama-8B address my main concerns. I will raise my score.

---

> > > ### Author Response · Authors · 2026-04-06
> > >
> > > Thank you for the time and effort you have devoted to reviewing our paper. We will carefully incorporate your suggestions into the next version of the paper.
> > >
> > > Best regards.

---

### Official Review · Reviewer_zXPC · 2026-03-12

**Soundness:** 2
**Presentation:** 3
**Significance:** 2
**Originality:** 2
**Overall Recommendation:** 4
**Confidence:** 3

**Summary:**

This paper addresses the challenge of fine-tuning reasoning models (e.g., Qwen3-8B) using synthetic data generated by stronger teacher models (e.g., GPT-OSS-120B). The authors identify "stylistic divergence" between teacher and student models as a primary cause for performance degradation during Supervised Fine-Tuning (SFT). To mitigate this, they propose TESSY (Teacher-Student Cooperation Data Synthesis), a framework that interleaves the two models to generate responses. Specifically, the teacher model generates "capability tokens" (reasoning content), while the student model generates "style tokens" (transitional text), guided by trained boundary predictors and a generate-then-rollback strategy. Experiments on code generation tasks show that TESSY improves performance on benchmarks like LiveCodeBench and OJBench where traditional teacher-only SFT leads to declines.

**Compliance With Llm Reviewing Policy:**

Affirmed.

**Final Justification:**

The authors have addressed all my concerns well, so I have adjusted my overall score to support the acceptance of this paper.

**Key Questions For Authors:**

- Given that specific "style" tokens (like "Wait") are known to peak during information-heavy reasoning steps, how does TESSY ensure that forcing the student to generate these doesn't lower the "reasoning floor" of the teacher's capability spans?

- How do the boundary predictors handle "edge cases" where reasoning and style are linguistically intertwined? A failure in the boundary predictor (Line 12 of Algorithm 1) could lead to "semantic inconsistencies". Have you measured the error rate of the 0.6B boundary model?

- Why does TESSY show such a marked improvement in code (OJBench) but fail to significantly move the needle on GPQA or OlympiadBench? Does this suggest the "style" conflict is primarily a formatting/syntax issue specific to code?

**Limitations:**

Yes

**Strengths And Weaknesses:**

**Strengths:**

- The paper is generally well-structured, providing clear algorithmic details and helpful visualizations of data distribution shifts.

**Weakness:**

- The reliance on binary classification for "capability" vs. "style" tokens is conceptually shaky. The authors acknowledge that tokens like "Wait" or "Hmm"—traditionally seen as style—may actually facilitate reasoning. This suggests that separating the two into distinct model-driven spans might disrupt the very reasoning process they aim to preserve. Furthermore, the boundary predictors are trained on only 100k samples using a Qwen3-0.6B-Base model, raising concerns about the accuracy and robustness of these predictors in complex, multi-step reasoning scenarios.

- While the results on code generation are positive, the significance is limited by the specialized nature of the task. On out-of-domain mathematical and science benchmarks (GPQA, OlympiadBench), TESSY provides negligible gains or merely maintains the baseline, suggesting the framework may not be the "flagship" solution for reasoning models the authors claim it to be.

- The concept of alternating generation or student-teacher collaboration is not entirely new, with the authors themselves citing similar works like AdaSwitch. The "generate-then-rollback" strategy is a relatively incremental engineering solution to the boundary problem.

---

> ### Author Rebuttal · Authors · 2026-03-31
>
> We sincerely appreciate the reviewer’s comments and the time and effort invested in reviewing our work. However, there are a few misunderstandings that we would like to clarify through discussion.
>
> ## Q1 of Reviewer zXPC. How does TESSY ensure that forcing the student to generate these doesn't lower the "reasoning floor" of the teacher's capability spans?
> Is the concern that our method might harm the teacher model’s generation quality?
>
> **As discussed in Figure 6 of our submitted paper, under the 40K length constraint (Qwen3-8B default), TESSY’s generation quality exceeds Teacher-Only by 10.99%. Increasing the Teacher-Only model’s maximum length to 64K improves its generation quality by only 1.72% compared to TESSY at 40K**. Although discourse markers like “Hmm” may affect reasoning, the teacher’s greater capability and robustness make some differences in generation patterns acceptable.
>
> The primary challenge for SFT reasoning models is catastrophic forgetting due to data distribution shifts. Even a high-quality teacher cannot benefit the student until this issue is addressed.
>
> ## Q2 of Reviewer zXPC. Concerns about the predictor.
> | |LCB-V5 |LCB-V6 |LCB-Pro |OJBench |
> |----|:----:|:---:|:------:|:-----:|
> | Qwen3-8B | 55.09 | 49.58 | 25.35  | 18.75 |
> | + Teacher-Only  | 41.32 | 36.57 | 22.10  |  8.73 |
> | + TESSY (Predictor trained on 500 samples) | 59.58 | 54.00 | 35.13  | 23.98 |
> | + TESSY (Predictor trained on 100k samples)| 60.68 | 55.15 | 35.84  | 26.51 |
>
> To address this concern, we trained the predictor on only 500 samples, regenerated the data, and compared it with data from the 100K-sample predictor.
> Evaluating the same 40K questions (see table above), the 500-sample predictor still far outperforms Teacher-Only.
>
> Furthermore, **the results in ``Q1 of Reviewer QVvU`` and ``Q3 of Reviewer QVvU``, as well as ``Q1 of Reviewer nTeU``, covering three tasks or student models, were all obtained using the predictor trained on 500 code samples, and all results are positive, showing it generalizes across tasks and models.**
>
> We evaluated the predictor’s boundary accuracy and found that 69.94% are correctly predicted using only 500 training samples. The 100K samples in the paper were an over-provisioned setting, and synthesizing and training on them is not much more costly than using 500 samples. Identifying separator tokens is simple, as memorizing common trigger words suffices for most patterns.
>
> ## Q3 of Reviewer zXPC. Why does TESSY show such a marked improvement in code but fail to significantly move the needle on GPQA or OlympiadBench?
> **Because the model is trained entirely on programming contest code, without any math-related data, it is unreasonable to expect the same level of improvement on math benchmarks as on code tasks.** The purpose of showing math and scientific reasoning performance in Table 1 is to evaluate whether different SFT methods cause catastrophic forgetting on tasks outside the training targets.
>
> Importantly, the gains of our method on math reasoning are not negligible. Qwen3-8B has been extensively trained, even overfitted on open-source math datasets (see Q1 of Reviewer nTeU), so further improvement is non-trivial. Despite not using any math data, our method achieves a 3.75% improvement on AIME2024 (averaged over 32 runs).
> For reference, a recent work (https://.org/pdf/2601.18734) reports that training on 30K math examples only yields about a 2% improvement on AIME2024.
>
> We understand that the reviewer may have concerns about whether our method can generalize to other reasoning tasks.
> Please refer to our response to ``Q1 of Reviewer nTeU``. We have added experiments using math and science reasoning datasets, and the results demonstrate that our method can indeed generalize to both math and science reasoning tasks.
>
> ## Q4 of Reviewer zXPC. The novelty of our work.
> Technical Advantages: **Unlike methods such as AdaSwitch, which require models from the same family and with the same tokenizer, our approach can synthesize data across different models.** Additionally, distinguishing and leveraging style and capability tokens in reasoning models sets our approach apart from existing studies.
>
> Motivation: Our work addresses the distribution shift problem in SFT for reasoning models, which remains underexplored in current research. In contrast, methods like AdaSwitch are evaluated on non-reasoning models and on traditional benchmarks such as GSM8K.
>
> Practical Objective: Our goal is to synthesize data that can be openly shared, whereas methods like AdaSwitch are not designed for generating shareable synthetic datasets.
>
> ## We look forward to your feedback
> We sincerely thank you again for your review. If there are any concerns we have not yet addressed, please do let us know. We will make every effort to respond to all your questions and hope that you might reconsider your evaluation of our work accordingly.

---

> > ### Author Rebuttal · Reviewer_zXPC · 2026-04-04
> >
> > The authors have addressed my concerns.

---

> > > ### Author Response · Authors · 2026-04-04
> > >
> > > Thank you for your response and for the time you have devoted to reviewing.
> > >
> > > In our previous rebuttal, we addressed your concerns regarding teacher generation quality, predictor robustness, generalization to other tasks, and the novelty of our work.
> > >
> > > **We are glad that our responses have adequately addressed your concerns, and that no additional concerns were raised.**
> > >
> > > However, we notice that the overall recommendation remains a weak reject. We would sincerely appreciate any further clarification on the concerns that may have led to this rating, so that we can better address them.
> > >
> > > Training models in an on-policy manner has recently attracted increasing attention. In particular, recent work has widely discussed self-distillation as a method to mitigate distribution mismatches. Our work provides a timely contribution by proposing a new approach to this problem, showing improvements over self-distillation methods on reasoning tasks (see Q1 of Reviewer dhnr).
> > >
> > > While we have made our best effort, we understand that our work may still have limitations that we have not fully identified. We sincerely respect your evaluation and rating, and if possible, we would greatly appreciate the opportunity to better understand any remaining concerns and address them during the remaining discussion period.
> > >
> > > Thank you again for your feedback!

---

### Decision · Program_Chairs · 2026-04-30

**Decision:**

Accept (regular)

**Comment:**

This paper proposes a framework for improving distillation-based fine-tuning by aligning teacher-generated reasoning traces with the student model’s style. The problem is timely and important, and reviewers find the approach intuitive, well-motivated, and clearly presented. The method demonstrates consistent empirical gains across coding benchmarks, and additional results in the rebuttal strengthen evidence of generalization beyond code and across model families. Overall, reviewers agree that the paper provides a practical and useful contribution.

That said, the contribution is somewhat incremental in novelty, as it builds on existing ideas in distillation and style alignment, and the work lacks theoretical analysis to more deeply justify the approach. Some concerns were also raised about baseline coverage and initial evaluation scope, though these were largely addressed during rebuttal.

In summary, while the paper does not introduce fundamentally new theoretical insights, it presents a simple, effective, and well-validated method for an important practical problem, which reviewers find valuable to the community.